# Generalization Performance of Ensemble Clustering: From Theory to Algorithm

**Xu Zhang** [1]  **Haoye Qiu** [1]  **Weixuan Liang** [2]  **Hui Liu** [3]  **Junhui Hou** [4]  **Yuheng Jia** [1 3 5]

## Abstract

Ensemble clustering has demonstrated great success in practice; however, its theoretical foundations remain underexplored. This paper examines the generalization performance of ensemble clustering, focusing on generalization error, excess risk and consistency. We derive a convergence rate of generalization error bound and excess risk bound both of $\mathcal{O}(\sqrt{\frac{\log n}{m}} + \frac{1}{\sqrt{n}})$, with $n$ and $m$ being the numbers of samples and base clusterings. Based on this, we prove that when $m$ and $n$ approach infinity and $m$ is significantly larger than $\log n$, i.e., $m, n \to \infty, m \gg \log n$, ensemble clustering is consistent. Furthermore, recognizing that $n$ and $m$ are finite in practice, the generalization error cannot be reduced to zero. Thus, by assigning varying weights to finite clusterings, we minimize the error between the empirical average clusterings and their expectation. From this, we theoretically demonstrate that to achieve better clustering performance, we should minimize the deviation (bias) of base clustering from its expectation and maximize the differences (diversity) among various base clusterings. Additionally, we derive that maximizing diversity is nearly equivalent to a robust (min-max) optimization model. Finally, we instantiate our theory to develop a new ensemble clustering algorithm. Compared with SOTA methods, our approach achieves average improvements of 6.0%, 7.3%, and 6.0% on 10 datasets w.r.t. NMI, ARI, and Purity. The code is available at https://github.com/xuz2019/GPEC.

---

[1]School of Computer Science and Engineering, Southeast University, Nanjing 210096, China [2]College of Computer Science and Technology, National University of Defense Technology, Changsha, China [3]School of Computing Information Sciences, Saint Francis University, Hong Kong, China [4]Department of Computer Science, City University of Hong Kong, Hong Kong, China [5]Key Laboratory of New Generation Artificial Intelligence Technology and Its Interdisciplinary Applications (Southeast University), Ministry of Education, China. Correspondence to: Yuheng Jia <yhjia@seu.edu.cn>.

*Proceedings of the 42$^{nd}$ International Conference on Machine Learning*, Vancouver, Canada. PMLR 267, 2025. Copyright 2025 by the author(s).

## 1. Introduction

Ensemble clustering has attracted great attention in recent years due to its high accuracy and robustness compared to single clustering algorithm. It integrates multiple clustering results to obtain a consensus one instead of the access to the original features of the data, making it broadly applicable across various scenarios (Strehl & Ghosh, 2002). Many scholars have made considerable efforts in this area. For example, Fred and Jain (Fred & Jain, 2005) utilized a voting mechanism to generate an $n \times n$ similarity matrix to describe the relationships between sample pairs ($n$ is the number of samples), and applied hierarchical clustering to derive the final clustering results. Huang (Huang et al., 2018) realized that the importance of clusters in ensemble pool varies and assigned different weights to various clusters by estimating their uncertainty. Recently, Jia (Jia et al., 2024) utilized the high-confidence relationships to propagate similarity and designed a self-enhancement framework for the similarity matrix. More researches on ensemble clustering can be found in (Topchy et al., 2005; Jia et al., 2019; Yi et al., 2012; Jia et al., 2021; Zhang, 2022; Zhou et al., 2024; Xu et al., 2024; Li & Jia, 2025; Peng et al., 2023).

Despite significant advances in practice, the theoretical analysis of ensemble clustering remains far from satisfactory. Theoretical analysis of an algorithm helps us understand its generalization performance such as generalization error, excess risk and consistency. Generalization error represents the expected loss of an algorithm across the entire data distribution. Excess risk refers to the difference between the expected loss of a model and the expected loss of the optimal model. For consistency, it means that whether a learning algorithm can uncover the true underlying structure of the data as the amount of training data increases. Most previous studies (Pollard, 1981; Bachem et al., 2017; Li et al., 2023) focus on the generalization performance of a single clustering algorithm. To the best of our knowledge, only one paper (Liu et al., 2017) has established a generalization error bound in the field of ensemble clustering while the excess risk and consistency are neglected. It demonstrates, from the perspective of weighted kernel $k$-means, that the generalization error bound of ensemble clustering is $\mathcal{O}(1/\sqrt{n})$. However, this work fails to consider that each base clustering should be treated as a random variable, which makes this study fundamentally no different

from the researches of the generalization error of a single clustering algorithm. In ensemble clustering, we should consider not only the distribution of the data but also the distribution of the base clusterings. This underscores the need to understand the relationship between the number of samples $n$ and the number of base clusterings $m$. Therefore, in this paper, we investigate the generalization error bound and excess risk bound for ensemble clustering, and get the conclusion that both of them are $\mathcal{O}(\sqrt{\frac{\log n}{m}} + \frac{1}{\sqrt{n}})$. Based on these results, we derive the sufficient conditions for the consistency of ensemble clustering: both $m$ and $n$ approach infinity and $m$ is significantly larger than $\log n$, i.e., $m, n \to \infty, m \gg \log n$.

Although the above conclusion reveals the relationship between $m$ and $n$ in ensemble clustering, it is impractical in the real world to actually acquire infinite sample points and base clusterings. Therefore, we further consider whether it is possible to reduce the loss between the empirical average of base clusterings and the expectation of base clustering. By deriving the loss function between them, we reveal that minimizing the deviation of each base clustering with its expectation (bias) and maximizing the differences among various base clusterings (diversity) can promote the clustering performance. However, once the base clusterings are given, both the bias and diversity are fixed. We, therefore, transform ensemble clustering into a learnable problem by weighting the base clusterings to decrease the loss, from which we also find that maximizing diversity is nearly equivalent to a robust optimization problem. By instantiating our theory, we design a new ensemble clustering algorithm and optimize it by the reduced gradient descent method. In summary, the key contributions of this work are:

- We pioneer the derivation of the generalization error bound and excess risk bound for ensemble clustering, incorporating considerations of both data and clustering distributions. We also establish sufficient conditions for the consistency of ensemble clustering, a novel advancement in the field.

- Our theoretical exploration uncovers that in ensemble clustering, minimizing bias between each base clustering and its expectation, alongside maximizing diversity among base clusterings, enhances clustering performance. Moreover, we establish a fundamental link between diversity and robustness in this context.

- Building upon our theoretical framework, we introduce a novel ensemble clustering algorithm and address it through the reduced gradient descent method, offering a practical solution based on rigorous theoretical underpinnings.

- By extensive experimental validation, we confirm the validity of our theoretical assertions and demonstrate

that the proposed algorithm surpasses other state-of-the-art methods significantly in terms of performance.

## 2. Preliminaries

In this section, we first briefly introduce key notations and general assumptions. Some of these align with (Von Luxburg et al., 2008) and (Liang et al., 2023), where readers can consult for further details. We then proceed to describe the co-association matrix in ensemble clustering.

### 2.1. Notations and General Assumptions

In this paper, let $n$ represent the number of samples and $m$ the number of base clusterings. $(\cdot)^{\top}$ and $\mathrm{tr}(\cdot)$ are used to transpose and calculate the trace of a matrix. $||\mathbf{A}||_{\mathrm{F}}$ is the Frobenius norm of a matrix. $||\mathbf{A}||_2$ donates the spectral norm of a matrix $\mathbf{A}$, $||\mathbf{a}||_2$ is $\ell_2$-norm for vector $\mathbf{a}$. $\mathbf{A} \preceq \mathbf{B}$ means $\mathbf{B} - \mathbf{A}$ is positive semi-definite.

We assume the sample space $\mathcal{X}$ is compact. Let $\rho(x)$ and $\rho_n(x)$ denote the corresponding true probability distribution and empirical distribution of $x$, respectively. The dataset $\mathrm{S}_n = \{x_1, \cdots, x_n\}$ is collected independently and identically distributed (i.i.d.) from $\mathcal{X}$ according to the distribution $\rho$. We denote $\pi^{(t)}$ as a base clustering generated i.i.d. by a clustering algorithm. $\pi^{(t)}(x_i)$ is the clustering label of the $t$-th base clustering for data $x_i$. We denote $\pi^{(t)}$ as an $n \times 1$ vector and $k^{(t)}$ as the number of clusters for $\pi^{(t)}$. $\Pi = \{\pi^{(1)}, \cdots, \pi^{(m)}\}$ is the ensemble base clustering pool with $m$ base clusterings.

### 2.2. Co-Association Matrix

In clustering, as no supervision is available, the labels we obtain are not aligned with the true labels of the samples. Nonetheless, the similarity relationship between sample pair is unique, we can define the similarity for each base clustering $\pi^{(t)}$ uniquely as

$$\mathbf{A}_{ij}^{(t)} = \delta(\pi^{(t)}(x_i), \pi^{(t)}(x_j)), \; \delta(a,b) = \begin{cases} 1, & \text{if } a = b, \\ 0, & \text{else.} \end{cases}$$

The CA matrix $\bar{\mathbf{A}}$ (Fred & Jain, 2005) is the average of these similarity matrices, $\bar{\mathbf{A}} = \frac{1}{m} \sum_{t=1}^{m} \mathbf{A}^{(t)}$. Since each similarity matrix $\mathbf{A}^{(t)}$ is a positive semi-definite matrix and $\bar{\mathbf{A}}$ is a convex combination of these matrices, the CA matrix is also positive semi-definite. CA-based ensemble clustering methods (Huang et al., 2016; 2021; Zhou et al., 2023; Ji et al., 2024) try to learn a more accurate CA matrix, and then perform hierarchical clustering or spectral clustering on it to obtain a more accurate consensus result.

## 3. Generalization Performance

Based on the definition in Section 2.2, we define the degree normalized similarity matrix $\mathbf{K}^{(t)}$ of $\mathbf{A}^{(t)}$ is $\mathbf{K}^{(t)} =$

$\mathbf{D}^{(t)-1/2}\mathbf{A}^{(t)}\mathbf{D}^{(t)-1/2}$ where $\mathbf{D}^{(t)-1/2}$ is the degree matrix of $\mathbf{A}^{(t)}$. Obviously $\mathbf{K}^{(t)}$ is still symmetric and positive semi-definite and we assume $\mathbf{K}^{(t)}$ is generated from a kernel function $K^{(t)}$, where $\mathbf{K}_{ij}^{(t)} = K^{(t)}(x_i, x_j)$. The empirical error function $\hat{F}\left(\hat{\mathbf{Z}}; \bar{\mathbf{K}}\right)$ for ensemble clustering is defined as:

$$\hat{F}\left(\hat{\mathbf{Z}}; \bar{\mathbf{K}}\right) = \frac{1}{n}\max_{\hat{\mathbf{Z}}\in\mathbb{R}^{n\times k}} \text{tr}\left(\hat{\mathbf{Z}}^\top\bar{\mathbf{K}}\hat{\mathbf{Z}}\right), \text{s.t. } \hat{\mathbf{Z}}^\top\hat{\mathbf{Z}} = \mathbf{I}, \quad (1)$$

where $\hat{\mathbf{Z}}$ represents the spectral embedding of (normalized) CA matrix $\bar{\mathbf{K}}$, which is utilized to approximate the cluster indicator matrix. $\bar{\mathbf{K}} = \frac{1}{m}\sum_{t=1}^m \mathbf{K}^{(t)}$ is the average of normalized similarity matrices, and the coefficient $\frac{1}{n}$ in Eq. (1) guarantees the convergence of eigenvalues of the kernel matrix to those of the corresponding integral operator as $n\to\infty$ (Liang et al., 2024; Rosasco et al., 2010). Let $\{\hat{\lambda}_q\}_{q=1}^k$ be the largest $k$ eigenvalues of $\frac{1}{n}\bar{\mathbf{K}}$. The solution to Eq. (1) is the eigenvectors $\hat{\mathbf{Z}} = [\mathbf{z}_1, \cdots, \mathbf{z}_k]$ corresponding to $k$ largest eigenvalues of $\bar{\mathbf{K}}$. Considering the true continuous distribution of the data, we define the following integral operator $L_K g(x): L^2(\mathcal{X}, \rho) \to L^2(\mathcal{X}, \rho)$

$$L_K g(x) = \int_{\mathcal{X}} K(x, y) g(y)\mathrm{d}\rho(y),$$

where $L^2$ denotes square-integrable function space. According to the definition of eigenfunction, we have

$$\zeta_q(x) = \frac{1}{\lambda_p}\int_{\mathcal{X}} K(x, y)\zeta_q(y)\mathrm{d}\rho(y),$$

where $\zeta_q(x)$ is the corresponding eigenfunction of $\lambda_q$, and $\lambda_q$ is the eigenvalue of $L_K$. Thus, we define the error measured over the entire distributions of data and base clusterings, referred to as the population-level error with the expectation of base clustering,

$$F(\mathcal{Z}; K^*) =$$
$$\max_{\{\zeta_q\}_{q=1}^k \in \Gamma} \sum_{q=1}^k \iint_{\mathcal{X}} K^*(x, y)\zeta_q(x)\zeta_q(y)\mathrm{d}\rho(x)\mathrm{d}\rho(y), \quad (2)$$

where $\mathcal{Z} = \{\zeta_q\}_{q=1}^k$ denotes the corresponding eigenfunctions of integral operator $L_{K^*}$ with eigenvalues $\{\lambda_q\}_{q=1}^k$. $K^*(x, y) = \mathbb{E}[K^{(t)}(x, y)]$ is the expectation of the normalized similarity function $K^{(t)}$. Note that $\mathbb{E}[\bar{K}] = \mathbb{E}[K^{(t)}] = K^*$, meaning the expectation of the CA function ($\bar{K}$) is the same as that of a single normalized similarity function. In the following sections, we will sometimes refer to $K^*$ as the expectation of the CA function.

However, as $\hat{\mathbf{Z}}$ and $\mathcal{Z}$ lie in the different space, we define the empirical integral operator, which is the approximation of the theoretical integral operator based on finite samples,

$\hat{L}_K: L^2(\mathcal{X}, \rho_n) \to L^2(\mathcal{X}, \rho_n)$ as

$$\hat{L}_K\hat{z}_q(x) = \frac{1}{n}\sum_{i=1}^n K(x, x_i)\hat{z}_q(x_i).$$

According to (Bengio et al., 2004), the eigenvalues of $\frac{1}{n}\bar{\mathbf{K}}$ and $\hat{L}_{\bar{K}}$ are the same except zero eigenvalues, and the empirical eigenfunctions of $\frac{1}{n}\bar{\mathbf{K}}$ are

$$\hat{z}_q(x) = \frac{1}{n\hat{\lambda}_q}\sum_{i=1}^n \bar{K}(x, x_i)\hat{z}_q(x_i),$$

where $\hat{z}_q(x_i) = \sqrt{n}\mathbf{z}_{iq}$. Thus, Eq. (1) is rewritten as

$$\hat{F}(\hat{Z}; \bar{K}) = \max_{\{\hat{z}_q\}_{q=1}^k} \frac{1}{n^2}\sum_{q=1}^k\sum_{i=1}^n\sum_{j=1}^n \bar{K}(x_i, x_j)\hat{z}_q(x_i)\hat{z}_q(x_j).$$

**Key problems**: According to the above definitions, we investigate the generalization performance of ensemble clustering including generalization error bound, excess risk bound, and sufficient conditions for consistency, which are defined as follows:

- Generalization error: the difference between empirical error and population-level error, represented as $\hat{F}(\hat{Z}; \bar{K}) - F(\mathcal{Z}; K^*)$;
- Excess risk: quantifying the difference in error between a learning algorithm and the optimal algorithm on data distribution, expressed as $F(\hat{Z}; K^*) - F(\mathcal{Z}; K^*)$;
- Consistency: the clusterings produced by the given algorithm converge to a clustering that represents the entire underlying space. That is, as the number of samples $n$ and base clusterings $m$ increase, the empirical eigenvectors $\hat{\mathbf{Z}}$ converge to the eigenfunctions $\mathcal{Z}$ of the true underlying structure.

The following three theorems address the key problems.

**Theorem 3.1.** *Under the general assumptions and assume that the gap between the $k$-th and $(k+1)$-th eigenvalues of the expectation of normalized similarity matrix $\mathbf{K}^*$ is $\delta_k$ and $\delta_k \geq \frac{1}{c} > 0$ where $c$ is a constant. For any $0 < \delta < 1$, with probability at least $1 - \delta$, we have*

$$\hat{F}\left(\hat{Z}; \bar{K}\right) - F\left(\mathcal{Z}; K^*\right) \quad (3)$$

$$\leq \left(2\sqrt{2}c + 1\right)\left(\frac{2}{3m}\log\frac{6n}{\delta} + \sqrt{\frac{8}{m}\log\frac{6n}{\delta}}\right) + \frac{2\sqrt{2}\log\left(\frac{6}{\delta}\right)}{\sqrt{n}}.$$

*Proof.* See Appendix A.1.1. □

**Remark**. Theorem 3.1, for the first time, presents the generalization error bound of ensemble clustering under the consideration of both the data distribution and the base clustering distribution, which is $\mathcal{O}(\sqrt{\frac{\log n}{m}} + \frac{1}{\sqrt{n}})$. Through this theorem, we establish the relationship between the sample size $n$ and the number of base clusterings $m$. Clearly, if

sample size $n$ is fixed, the generalization error continues to decrease as the number of base clusterings increases, although it will not converge to zero. However, with a fixed $m$, we cannot guarantee the decrease of generalization error, instead, it tends to infinity as $n$ increases. Thus, *in ensemble clustering, simply acquiring more samples is not an effective strategy, we still need to obtain more base clusterings as data size increases*. Additionally, a rapid growth of $m$ is required to allow the generalization error to converge to $0$, which implies that $m$ should be significantly larger than $\log n$ (i.e., $m \gg \log n$). In practice, we recommend setting $m = \sqrt{n}$ to strike a balance between theoretical convergence and computational efficiency of time and space.

**Theorem 3.2.** *Under the same assumptions as Theorem 3.1 and with the additional condition that $||\hat{z}_q||_\infty \leq \sqrt{c_0}$ ($c_0 > 0$ is a constant), for any $0 < \delta < 1$, with probability at least $1 - \delta$, we have*

$$F\left(\hat{Z}; K^*\right) - F\left(\mathcal{Z}; K^*\right) \leq k\left(\frac{2\sqrt{2}c_0}{\sqrt{n}} + \sqrt{\frac{8\log\frac{3}{\delta}}{n}}\right)$$

$$+ 2\sqrt{2}c\left(\frac{2}{3m}\log\frac{6n}{\delta} + \sqrt{\frac{8}{m}\log\frac{6n}{\delta}}\right) + \frac{2\sqrt{2}\log\left(\frac{6}{\delta}\right)}{\sqrt{n}}. \quad (4)$$

*Proof.* See Appendix A.1.2. □

**Remark**. Theorem 3.2 provides the excess risk bound for ensemble clustering, which is also expressed as $\mathcal{O}(\sqrt{\frac{\log n}{m}} + \frac{1}{\sqrt{n}})$. Obviously, we require the same condition as in Theorem 3.1 (i.e., $m, n \to \infty, m \gg \log n$) to ensure that the population-level error ($F(\hat{Z}; K^*)$) of the learned algorithm ($\hat{Z}$) converges to that ($F(\mathcal{Z}; K^*)$) of the optimal algorithm ($\mathcal{Z}$) on the entire data and clustering distributions. It is worth noting that this theorem introduces an additional mild assumption $||\hat{z}_q||_\infty \leq \sqrt{c_0}$, which is easily satisfied given that $\bar{K}(x, x_i) \leq 1$, $\sum_{i=1}^n \hat{z}_q(x_i) \leq n$ and $\hat{z}_q(x) \leq \frac{1}{\hat{\lambda}_q}$.

**Theorem 3.3.** *Under the same assumptions as Theorem 3.1, if $m, n \to \infty$ and $\lim_{m,n\to\infty} \frac{\log n}{m} \to 0$, there exists a sequence $(a_q)_q \in \{-1, 1\}$ such that*

$$\|a_q\hat{\mathbf{z}}_q - \zeta_q\|_\infty \to 0,$$

*in probability.*

*Proof.* See Appendix A.1.3. □

**Remark**. Theorem 3.3 provides the sufficient conditions for the consistency of ensemble clustering. It describes that the corresponding empirical eigenvectors converge to the eigenfunctions in the limit case. Based on this, we conclude that the clustering learned from empirical data can converge to the true underlying structure of the data, thereby ensuring the consistency of ensemble clustering. Note that

since multiplying the eigenvectors by $\pm 1$ does not affect the outcome, we need to prepend a coefficient $a_q$ to $\hat{\mathbf{z}}_q$ to ensure that the signs of $\hat{\mathbf{z}}_q$ and $\zeta_q$ are consistent.

# 4. Key Factors in Ensemble Clustering

While the preceding section offers theoretical guarantees for the performance of ensemble clustering when both $m$ and $n$ approach infinity, and explores the relationship between $m$ and $n$, it is not feasible to obtain infinite data points and base clusterings in practice. Therefore, in this section, we consider how to approximate the expectation of clustering ($K^*$) with the average of the finite base clusterings (the CA matrix $\bar{K}$) by the following optimization problem

$$\min \mathcal{L} = \hat{F}\left(\hat{\mathbf{Z}}; \bar{\mathbf{K}}\right) - \hat{F}\left(\hat{\mathcal{Z}}; K^*\right)$$
$$= \frac{1}{n}\text{tr}\left(\hat{\mathbf{Z}}^\top \bar{\mathbf{K}}\hat{\mathbf{Z}}\right) - \frac{1}{n}\text{tr}\left(\hat{\mathcal{Z}}^\top K^*\hat{\mathcal{Z}}\right). \quad (5)$$

When $\mathcal{L} = 0$, we perfectly fit the underlying structure of the samples using a finite number of base clusterings. However, once the base clusterings are established, $\bar{K}$ is fixed and so as to the associated $\mathcal{L}$. To decrease the loss $\mathcal{L}$, we apply different weights to various base clusterings. Accordingly, we substitute CA matrix $\bar{K}$ with weighted CA matrix $\mathbf{K}^{\mathbf{w}}$, which is defined as

$$\mathbf{K}^{\mathbf{w}} = \sum_{t=1}^m w_t \mathbf{K}^{(t)}. \quad (6)$$

We replace $\mathcal{L}$ as $\mathcal{L}^{\mathbf{w}}$ and obtain the follow theorem.

**Theorem 4.1.** *Based on Eqs. (5) and (6) and let $c' = k/n + 2\sqrt{2}/\left(\lambda_k(K^*) - \lambda_{k+1}(K^*)\right)$, $\lambda_k(K^*)$ is the $k$-th eigenvalue of $K^*$, , $\tilde{w}_t = mw_t$, $m$ is the number of base clusterings, we derive the Bias-Diversity decomposition for ensemble clustering, as*

$$\min_{\mathbf{w}} \mathcal{L}^{\mathbf{w}} = \hat{F}\left(\hat{\mathbf{Z}}; \mathbf{K}^{\mathbf{w}}\right) - \hat{F}\left(\hat{\mathcal{Z}}; K^*\right) \quad (7)$$

$$\leq c'\sqrt{\frac{1}{m}(\underbrace{\sum_{t=1}^m \|\tilde{w}_t\mathbf{K}^{(t)} - K^*\|_F^2}_{\text{Bias}} - \underbrace{\sum_{t=1}^m \|\tilde{w}_t\mathbf{K}^{(t)} - \mathbf{K}^{\mathbf{w}}\|_F^2}_{\text{Diversity}})}.$$

*Proof.* See Appendix A.2.1. □

**Remark**. This theorem describes the loss $\mathcal{L}^{\mathbf{w}}$ is governed by two terms: Bias and Diversity. Here, Bias describes the average gap between each single weighted base clustering ($\tilde{w}_t\mathbf{K}^{(t)}$) and the expectation of base clustering ($K^*$), while Diversity describes the average difference between each single weighted base clustering ($\tilde{w}_t\mathbf{K}^{(t)}$) and the weighted CA matrix ($\mathbf{K}^{\mathbf{w}}$). Therefore, by adjusting $\mathbf{w} = \{w_t\}_{t=1}^m$ to achieve low Bias and high Diversity, we can reduce the loss $\mathcal{L}^{\mathbf{w}}$ and obtain better clustering performance.

To better analyze Theorem 4.1, we first simply Eq. (7) into a more concise from:

$$\min_{\mathbf{w}} \ -2\mathrm{tr}\left(\mathbf{K^w K^*}\right) + \mathrm{tr}\left(\mathbf{K^w K^w}\right)$$
$$\text{s.t. } \mathbf{w}^\top \mathbf{w} = 1, \mathbf{w} \geq 0. \tag{8}$$

where the constraint $\mathbf{w}^\top \mathbf{w} = 1$ is imposed to avoid sparse solutions for the weights $\mathbf{w}$. The proof of Eq. (7) $\Rightarrow$ Eq. (8) is provided in Appendix A.2.2. Eq. (8) remains a non-convex optimization problem of $\mathbf{w}$ and we still need to process $\mathbf{K^w}$ to obtain the final clustering results, such as performing hierarchical clustering or spectral clustering on it. To this end, we introduce the spectral embedding $\mathbf{Z}$ of $\mathbf{K^w}$ to Eq. (8). Specifically. the first term of Eq. (8) ($\min_{\mathbf{w}} -2\mathrm{tr}\left(\mathbf{K^w K^*}\right)$) is reformulated as $\max_{\mathbf{Z}} 2\mathrm{tr}(\mathbf{K^* ZZ}^\top)$ by substituting $\mathbf{K^w}$ for $\mathbf{ZZ}^\top$. For the second term $\min_{\mathbf{w}} \mathrm{tr}(\mathbf{K^w K^w})$, we replace one instance of $\mathbf{K^w}$ with the spectral embedding $\mathbf{Z}$, i.e., $\mathrm{tr}(\mathbf{K^w K^w}) \Rightarrow \max_{\mathbf{Z}} \mathrm{tr}(\mathbf{K^w ZZ}^\top)$, and further transform it into a min-max optimization problem. Besides, the original constraint $\mathbf{w}^\top \mathbf{w} = 1$ is non-convex, we revise it to $\mathbf{w}^\top \mathbf{1} = 1$ (where $\mathbf{1}$ is a column vector of all ones) and also modify the definition of $\mathbf{K^w} = \sum_{t=1}^{m} w_t^2 \mathbf{K}^{(t)}$, allowing $\mathbf{w}$ to be better interpreted as a weight distribution. Together with orthogonal constraint on the spectral embedding $\mathbf{Z}$, the optimization problem is finally redefined as:

$$\max_{\mathbf{Z} \in \mathbb{R}^{n \times k}} 2\mathrm{tr}\underbrace{\left(\mathbf{K^* ZZ}^\top\right)}_{-\text{Bias}} + \overbrace{\min_{\mathbf{w}} \max_{\mathbf{Z} \in \mathbb{R}^{n \times k}} \mathrm{tr}\underbrace{\left(\mathbf{K^w ZZ}^\top\right)}_{-\text{Diversity}}}^{\text{Robust optimization}}$$
$$\text{s.t. } \mathbf{Z}^\top \mathbf{Z} = \mathbf{I}, \mathbf{w}^\top \mathbf{1} = 1, \mathbf{w} \geq 0. \tag{9}$$

**Remark 1 (Diversity)**. From Eq. (9), we observe that the Diversity term aims to enhance the diversity among the base clusterings. Although some heuristic methods (Fern & Brodley, 2003; Kuncheva & Vetrov, 2006; Hadjitodorov et al., 2006; Jia et al., 2011; Metaxas et al., 2023) were proposed to increase diversity in ensemble clustering, our approach is entirely derived from Theorem 4.1, offering solid theoretical guarantees.

**Remark 2 (Robust Optimization)**. We surprisingly discover that maximizing the Diversity term is equivalent to a robust min-max optimization model (aiming to identify the spectral embedding that performs well even with a bad weight vector), which is similar to some existing robust ensemble algorithm (Liu, 2023; Zhang et al., 2022; Bang et al., 2018; Tao et al., 2019; Liang et al., 2022). Unlike their motivation to enhance the model's resistance to noise, *we explain that these algorithms essentially improve diversity within the ensemble to reduce the loss ($\mathcal{L}^{\mathbf{w}}$) between the empirical and expected error. This provides a theoretical explanations for why these algorithms work effectively.*

**Remark 3 (Bias)**. It is worth noting that existing algorithms (Bang et al., 2018; Liu, 2023) only consider the optimization of diversity (robustness), while neglecting the Bias term. A natural concern arises for those methods: *does min-max optimization sacrifice the most accurate individuals in the ensemble?* For example, if most individuals in the ensemble have high accuracy but a few have low accuracy, considering diversity might lead us to assign higher weights to the poorer performers, potentially dragging down the final consensus result (we will verify this in our experiments). Regrettably, existing algorithms neglect this issue. Our theory indicates that better clustering performance will be more likely to achieve by simultaneously optimizing (minimizing) bias and (maximizing) diversity in ensemble clustering.

## 5. Instantiation of Theorem 4.1

In this section, we instantiate our theoretical analysis (Eq. (9)) to obtain a novel ensemble clustering algorithm. Eq. (9) is not directly usable as we do not know the true expected value of the CA matrix $\mathbf{K^*}$. Therefore, we try to approximate it using a simple yet effective way.

### 5.1. Approximate $\mathbf{K^*}$

We extract the high-confidence elements in the CA matrix to approximate $\mathbf{K^*}$. This motivation is that if two samples belong to the same cluster, their pairwise value in CA matrix is more likely to be higher, which is reflected in the high-confidence elements of the CA matrix, as illustrated in Fig. 1. It is evident that as the values in the CA matrix increase, the precision of the corresponding elements also improves. Therefore, high-confidence elements from the CA matrix can well approximate the ground-truth relationship between two samples. Specifically, the high-confidence elements are calculated by

$$\mathbf{H}_{ij} = \begin{cases} \bar{\mathbf{K}}_{ij}, & \bar{\mathbf{K}}_{ij} \geq \alpha, \\ 0, & \text{else} \end{cases} . \tag{10}$$

where $\alpha$ is a hyper-parameter. Eq. (10) retains the high-confidence elements in the CA matrix and discards the low-confidence ones. However, $\mathbf{H}$ in Eq. (10) is generally very sparse (as illustrated in Fig. 1, the recall and proportion rates of the elements in $\mathbf{H}$ decrease as $\alpha$ increases) and not semi-positive definite. Therefore, we compute its second-order similarity relations to make it denser and semi-positive by

$$\tilde{\mathbf{K}} = \left(\mathbf{D}^{-1}\right)^\top \mathbf{H}^\top \mathbf{H} \mathbf{D}^{-1}, \tag{11}$$

where $\mathbf{D}$ is a diagonal matrix and $\mathbf{D}_{ii} = \sqrt{\sum_{i=1}^{n} \left(\mathbf{H}_{ij}\right)^2}$.

### 5.2. Proposed Ensemble Clustering Algorithm

After approximating $\mathbf{K^*}$ by $\tilde{\mathbf{K}}$ in Eq. (11), Eq. (9) can be instantiated into the following practical ensemble clustering

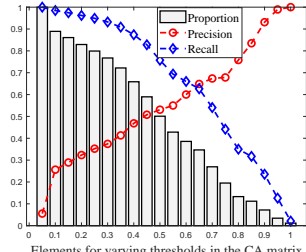 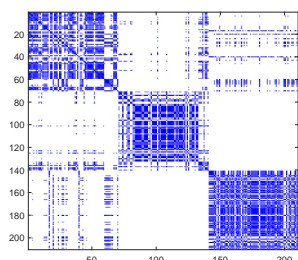 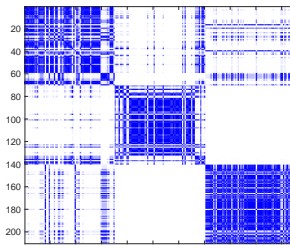 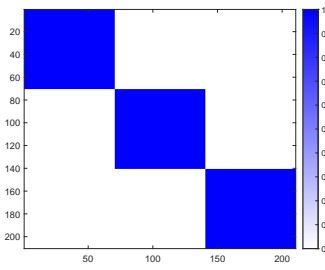

(a) The proportion, precision, and recall of high-confidence elements in different threshold.

(b) High confidence matrix $\mathbf{H}$ with threshold $\alpha = 0.4$.

(c) Second-order similarity relation matrix $\tilde{\mathbf{K}}$ of $\mathbf{H}$.

(d) The ground truth similarity matrix of seed dataset.

*Figure 1.* As shown in Fig. (a), with the increase in the high-confidence threshold, the proportion of elements and the recall rate gradually decrease, but the precision approaches 1, suggesting that high-confidence elements are reliable. Fig. (b) displays the visualization of the high-confidence matrix at a threshold of 0.4, which resembles the ground truth shown in Fig. (d), although it is still not dense enough. Consequently, we computed the second-order similarity relationship $\tilde{\mathbf{K}}$ of $\mathbf{H}$, as depicted in Fig. (c), which more closely approximates the ground truth (We use Seeds dataset for this experiment)

.

method (we provide brief proof of Eq. (9) $\Rightarrow$ Eq. (12) in Appendix A.2.3).

$$\min_{\mathbf{w}} \max_{\mathbf{Z}} \operatorname{tr}\left(\left(2\tilde{\mathbf{K}} + \mathbf{K}^{\mathbf{w}}\right) \mathbf{Z}\mathbf{Z}^\top\right)$$
$$\text{s.t. } \mathbf{Z}^\top \mathbf{Z} = \mathbf{I}, \sum_{t=1}^{m} w_t = 1, w_t \geq 0. \tag{12}$$

Since both $\tilde{\mathbf{K}}$ and $\mathbf{K}^{\mathbf{w}}$[1] are positive semi-definite, the problem is a convex problem with respect to $\mathbf{w}$. Theoretically, we can obtain the global minimum point for $\mathbf{w}$. Once the spectral embedding $\mathbf{Z}$ is obtained, we apply $k$-means algorithm to it to derive the final discrete clustering results.

### 5.3. Optimization of Eq. (12)

Eq. (12) is a typical min-max optimization problem with multi-variables. The usual approach is to fix one variable and optimize the other, but this approach often fails to yield a globally optimal solution. In (Liu, 2023), the author transformed this problem into minimizing the optimal value function and employed reduced gradient descent for solving it. Given the similarity of our problem-solving approach to this method, we provide only the key steps of the optimization process. Readers are encouraged to refer to (Liu, 2023) for more details.

For Eq. (12), we rewrite it as (the constraints have been omitted for brevity)

$$\min_{\mathbf{w}} \mathcal{J}(\mathbf{w}), \mathcal{J}(\mathbf{w}) = \left\{\max_{\mathbf{Z}} \operatorname{tr}\left(\left(2\tilde{\mathbf{K}} + \mathbf{K}^{\mathbf{w}}\right) \mathbf{Z}\mathbf{Z}^\top\right)\right\}.$$

---

[1]In this context, we adopt a cluster-weighting strategy to pre-assign weights to different clusters of the base clusterings, following approaches such as LWCA (Huang et al., 2018) and NWCA (Zhang et al., 2024). In this paper, we employ the NWCA method.

As established in (Bonnans & Shapiro, 1998), $\mathcal{J}(\mathbf{w})$ is differentiable,

$$\frac{\partial \mathcal{J}(\mathbf{w})}{\partial w_t} = 2w_t \operatorname{tr}\left(\mathbf{K}^{\mathbf{w}}\mathbf{Z}^*\mathbf{Z}^{*\top}\right),$$

where $\mathbf{Z}^* = \{\arg\max_{\mathbf{Z}} \operatorname{tr}((2\tilde{\mathbf{K}} + \mathbf{K}^{\mathbf{w}})\mathbf{Z}\mathbf{Z}^\top), \mathbf{Z}^\top\mathbf{Z} = \mathbf{I}\}$. Building upon this, we compute the gradient of $\mathcal{J}(\mathbf{w})$ as follows:

$$[\nabla \mathcal{J}(\mathbf{w})]_t = \frac{\partial \mathcal{J}(\mathbf{w})}{\partial w_t} - \frac{\partial \mathcal{J}(\mathbf{w})}{\partial w_u} \ \forall t \neq u,$$

and

$$[\nabla \mathcal{J}(\mathbf{w})]_u = \sum_{t=1, t\neq u}^{m} \frac{\partial \mathcal{J}(\mathbf{w})}{\partial w_u} - \frac{\partial \mathcal{J}(\mathbf{w})}{\partial w_t},$$

where $w_u$ is not selected as the zero component of $\mathbf{w}$. To address the constraint $\mathbf{w} \geq 0$, the final descent direction is computed as

$$d_t = \begin{cases} 0, & \text{if } w_t = 0 \text{ and } [\nabla \mathcal{J}(\mathbf{w})]_t > 0, \\ -[\nabla \mathcal{J}(\mathbf{w})]_t, & \text{if } w_t > 0 \text{ and } t \neq u, \\ -[\nabla \mathcal{J}(\mathbf{w})]_u, & \text{if } t = u, \end{cases} \tag{13}$$

where $d_t$ is the $p$-th component of gradient vector $\mathbf{d}$. We use gradient descent to set $\mathbf{w}_{t+1} \leftarrow \mathbf{w}_t + \beta\mathbf{d}$ and continue until the algorithm converges, where $\beta$ is a learning rate. The pseudo code for this algorithm is provided in Appendix C.

## 6. Experiments

### 6.1. Comparative Experiment

We evaluated our method on 10 datasets with method CEAM (Zhou et al., 2024), CEs²L, CEs²Q (Li et al., 2019), LWEA

*Table 1.* Performance (%) evaluation of different datasets based on the NMI metric. We have highlighted the values of the best-performing method in **bold**, and the second-best method is marked with an underline.

| Method | D1 | D2 | D3 | D4 | D5 | D6 | D7 | D8 | D9 | D10 | Average |
|---|---|---|---|---|---|---|---|---|---|---|---|
| CEAM (TKDE'24) | $5.6_{\pm10}$ | $36.2_{\pm26}$ | $16.7_{\pm4}$ | $27.4_{\pm1}$ | $60.1_{\pm10}$ | $4.3_{\pm3}$ | $18.0_{\pm2}$ | $19.0_{\pm5}$ | $14.3_{\pm4}$ | $8.8_{\pm5}$ | $21.0_{\pm8}$ |
| CEs²L (AIJ'19) | $3.4_{\pm5}$ | $9.3_{\pm10}$ | $19.0_{\pm4}$ | $27.9_{\pm2}$ | $45.1_{\pm14}$ | $12.3_{\pm5}$ | $12.0_{\pm2}$ | $15.2_{\pm7}$ | $\underline{15.7_{\pm3}}$ | $10.2_{\pm6}$ | $17.0_{\pm6}$ |
| CEs²Q (AIJ'19) | $2.5_{\pm4}$ | $11.5_{\pm8}$ | $17.6_{\pm5}$ | $28.1_{\pm3}$ | $43.9_{\pm15}$ | $12.1_{\pm5}$ | $12.2_{\pm2}$ | $17.9_{\pm4}$ | $15.4_{\pm3}$ | $7.5_{\pm4}$ | $16.9_{\pm6}$ |
| LWEA (TCYB'18) | $0.4_{\pm0}$ | $53.3_{\pm3}$ | $15.9_{\pm3}$ | $28.1_{\pm1}$ | $63.3_{\pm3}$ | $12.1_{\pm5}$ | $13.7_{\pm3}$ | $21.0_{\pm4}$ | $14.7_{\pm1}$ | $7.9_{\pm4}$ | $23.0_{\pm3}$ |
| NWCA (arXiv'24) | $0.4_{\pm0}$ | $52.5_{\pm3}$ | $16.0_{\pm3}$ | $28.4_{\pm1}$ | $63.7_{\pm3}$ | $12.5_{\pm4}$ | $13.6_{\pm3}$ | $21.7_{\pm1}$ | $14.8_{\pm1}$ | $9.7_{\pm4}$ | $23.3_{\pm2}$ |
| ECCMS (TNNLS'24) | $0.4_{\pm0}$ | $50.7_{\pm19}$ | $18.4_{\pm5}$ | $28.2_{\pm0}$ | $64.7_{\pm3}$ | $12.3_{\pm5}$ | $12.9_{\pm3}$ | $\underline{22.8_{\pm4}}$ | $15.5_{\pm2}$ | $9.1_{\pm4}$ | $23.5_{\pm5}$ |
| MKKM (arXiv'18) | $8.1_{\pm12}$ | $40.8_{\pm20}$ | $12.8_{\pm3}$ | $20.6_{\pm6}$ | $55.4_{\pm9}$ | $12.0_{\pm5}$ | $19.7_{\pm4}$ | $14.3_{\pm4}$ | $12.0_{\pm7}$ | $9.1_{\pm6}$ | $20.5_{\pm8}$ |
| SMKKM (TPAMI'23) | $8.7_{\pm4}$ | $38.5_{\pm11}$ | $19.3_{\pm4}$ | $27.0_{\pm2}$ | $59.4_{\pm9}$ | $10.5_{\pm5}$ | $\underline{20.0_{\pm2}}$ | $18.2_{\pm3}$ | $15.5_{\pm2}$ | $10.5_{\pm4}$ | $22.8_{\pm5}$ |
| SEC (TKDE'17) | $9.2_{\pm12}$ | $24.9_{\pm18}$ | $17.3_{\pm4}$ | $21.9_{\pm5}$ | $36.0_{\pm17}$ | $12.8_{\pm4}$ | $15.5_{\pm3}$ | $13.6_{\pm7}$ | $9.9_{\pm6}$ | $7.1_{\pm4}$ | $16.8_{\pm9}$ |
| Proposed ($\alpha = 0.1$) | $\underline{25.0_{\pm12}}$ | $\underline{58.3_{\pm1}}$ | $20.0_{\pm4}$ | $29.4_{\pm2}$ | $67.5_{\pm3}$ | $\underline{14.4_{\pm4}}$ | $18.8_{\pm2}$ | $19.6_{\pm6}$ | $15.0_{\pm4}$ | $\underline{12.4_{\pm4}}$ | $28.0_{\pm4}$ |
| Proposed | $\mathbf{25.0_{\pm12}}$ | $\mathbf{58.3_{\pm1}}$ | $\mathbf{21.1_{\pm3}}$ | $\mathbf{29.4_{\pm2}}$ | $\mathbf{67.5_{\pm3}}$ | $\mathbf{15.0_{\pm4}}$ | $\mathbf{22.9_{\pm2}}$ | $\mathbf{27.5_{\pm2}}$ | $\mathbf{15.8_{\pm3}}$ | $\mathbf{12.4_{\pm4}}$ | $\mathbf{29.5_{\pm4}}$ |

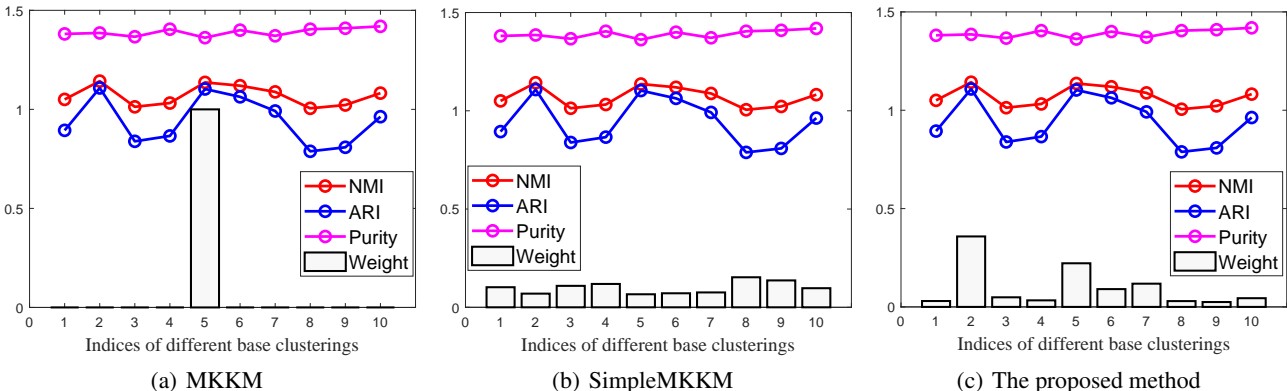

(a) MKKM        (b) SimpleMKKM        (c) The proposed method

*Figure 2.* Illustration of the performance (line plot, NMI, ARI, and Purity) of each individual base clustering when the ensemble size is 10, as well as the clustering weights learned by the three different methods (bar plot). Note that all three metrics are better when they possess higher values. For better visualization, we add 0.5 to the values of each metric.

(Huang et al., 2018), NWCA (Zhang et al., 2024), ECCMS (Jia et al., 2024), MKKM (Bang et al., 2018), SMKKM (Liu, 2023), SEC (Liu et al., 2017). Due to the space limitations, detailed descriptions of the datasets and comparison methods are provided in Appendix E.1 and E.2. For each dataset, we repeat the experiments 20 times and compute the average performance. The true number of clustering class is chosen as $k$ for each dataset. Three performance metrics are selected to evaluate the methods: NMI, ARI, and Purity, and larger value indicates better performance. Table 1 reports the comparisons based on the NMI metric, while the results for ARI and Purity are provided in Appendix E.3. As shown in Table 1, we observe that:

- The proposed method outperforms the comparison methods across all datasets. In terms of average performance, we exceed the second-best method by 6.0%, 7.3%, and 6.0% in NMI, ARI, and Purity, respectively.
- On some difficulty datasets, the performance advantage

of our method is more significant. For example, in the D1 (Phishing) dataset, the results obtained by other methods are close to 0 as measured by NMI, rendering them nearly impractical for guiding applications, while our method can provide some valuable information.

- Even with the hyper-parameter $\alpha = 0.1$ fixed, our method outperforms the methods compared in most datasets and lead on average across all methods. For example, with fixed hyper-parameter, we respectively lead the second-best method by 4.5%, 6.2%, and 4.4% in NMI, ARI, and Purity on average.

To further substantiate the efficacy of the modified method, we carry out hyper-parameter sensitivity experiment, ablation study, and ensemble size experiment. Due to space limitations, these experiments are detailed in Appendices E.4, E.5 and E.6.

*Table 2.* The detailed performance metrics of three different learning weight methods.

| Method | Objective Function | Essence | NMI | ARI | Puritty | Bias* | $-$Diversity* | Total* |
|---|---|---|---|---|---|---|---|---|
| MKKM | $\min_{\mathbf{w}} \min_{\mathbf{Z}} \operatorname{tr}\left(\mathbf{K^w}\left(\mathbf{I} - \mathbf{ZZ}^\top\right)\right)$ | min Diversity | 62.8 | 62.1 | 85.0 | **-94.6** | 116.2 | 21.6 |
| SMKKM* | $\min_{\mathbf{w}} \max_{\mathbf{Z}} \operatorname{tr}\left(\mathbf{K^w ZZ}^\top\right)$ | max Diversity | 65.3 | 66.2 | 85.9 | -10.0 | **1.0** | -9.0 |
| Proposed | $\max_{\mathbf{Z}} \operatorname{tr}\left(\mathbf{K}^* \mathbf{ZZ}^\top\right)$
$\min_{\mathbf{w}} \max_{\mathbf{Z}} \operatorname{tr}\left(\mathbf{K^w ZZ}^\top\right)$ | min Bias
max Diversity | **71.9** | **72.4** | **90.2** | -24.0 | 6.4 | **-17.6** |

\* "Bias" refers to $-2\operatorname{tr}(\mathbf{K^w K}^*)$, "$-$Diversity" is defined as $\operatorname{tr}(\mathbf{K^w K^w})$ and "Total" is equal to "Bias $-$ Diversity".

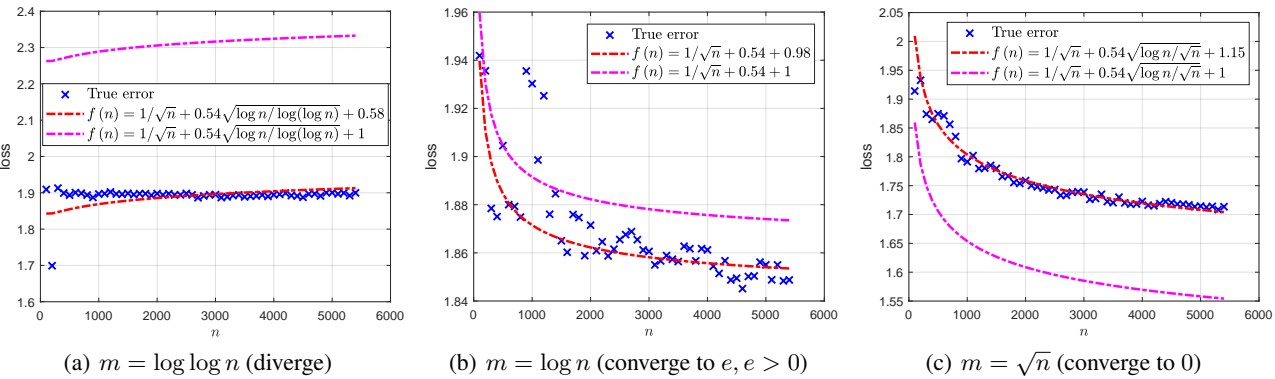

(a) $m = \log \log n$ (diverge)  (b) $m = \log n$ (converge to $e, e > 0$)  (c) $m = \sqrt{n}$ (converge to 0)

*Figure 3.* We conducted experiments on real data for Theorem 3.2. In this experiment, we uniformly sample data with an increment of 100 for the validation of $n$. The blue dots represent the errors computed from the real data, while the red and pink lines represent the fittings using the formulas from Theorem 3.2.

## 6.2. Clustering Weight Analysis

In this section, we analyze the base clustering weights learned by our method and compare them with those of other base clusterings weighted averaging method: MKKM (Bang et al., 2018), SimpleMKKM (SMKKM) (Liu, 2023). The details of these methods are summarized in Table 2, where we compare them across multiple metrics. As shown in Table 2, MKKM inherently reduces the diversity of base clusterings, thereby concentrating the weights on a single base clustering, as illustrated in Fig. 2. When the selected single base clustering aligns closely with the ground truth, MKKM significantly reduces bias, thereby enhancing clustering performance. However, this approach often faces two defects: 1) it does not always assign higher weight to the more accurate base clustering, as it lacks supervision; and 2) even if the best base clustering is selected every time, this method loses the advantage of ensemble learning that uses multiple base clusterings to achieve a better one. The objective function of the SimpleMKKM method is designed to enhance the diversity among base clusterings (as Table 2 reports), thereby distributing the weights as possible across multiple distinct base clusterings, as illustrated in Fig. 2. However, their weight values seem to follow an opposite trend to the performance of individual clustering, validat-

ing our earlier discussion (**Remark 3** in Section 4) that assigning weights solely to enhance diversity can lead to misallocation. Our method introduces high-confidence elements to guide the diversity and reduce bias, ensuring that higher weights are assigned to more accurate base clusterings. As a result, the proposed method achieves the lowest "Total" loss and enhances the final clustering performance. As a summary, the above analysis is consistent with our theoretical findings.

## 6.3. Validation of Excess Risk Bound

In this section, we validate Theorem 3.2 using the real dataset WFRN. Theorem 3.2 exhibits three distinct scenarios of excess risk: divergence, convergence to a constant greater than zero, and convergence to zero. We conduct experiments for these scenarios with $m = \log \log n$, $m = \log n$, and $m = \sqrt{n}$, respectively. Since we cannot obtain the expectation of the CA matrix, we substitute it with the similarity matrix produced by the ground-truth label. Therefore, the fitting function is defined as $a_1 \sqrt{n} + a_2 \sqrt{\log n / m} + \text{gap}$, where the gap represents the difference between the similarity matrix of the labels and the expected value of the CA matrix. It can be observed in Fig. 3 that the function fits the loss data points accurately when we choose $a_1 = 1$ and

$a_2 = 0.54$, as indicated by the red line. Theoretically, our gap should be a fixed value, as illustrated by the pink line. Although it does not completely conform to our data, the trend is consistent with the loss. From this experiment, we observe that when $m = \log \log n$, the loss value remains almost stable at 1.9; when $m = \log n$, the loss value shows a clear decreasing trend and eventually arrives at 1.85; and when $m = \sqrt{n}$, the loss curve exhibits a steady decline, ultimately reaching a loss value of approximately 1.7. Combining this experiment with our theoretical analysis, we further demonstrate that when $m \ll \log n$, the excess risk diverges; when $m = \mathcal{O}(\log n)$, it converges to a constant greater than 0; and when $m \gg \log n$, it converges to 0.

## 7. Conclusion and Discussion

In this paper, we have presented the generalization error bound, excess risk bound, and sufficient conditions for the consistency of ensemble clustering. Through this, we have elucidated the interplay between sample size $n$ and the number of base clustering $m$, offering insights relevant to practical applications. By approximating clustering expectations using weighted finite clustering, we identified the impact of Bias and Diversity on the errors between them. Notably, we have shown that maximizing Diversity aligns closely with robust optimization principles. Our contribution extends to the introduction of a novel ensemble clustering algorithm rooted in our theoretical framework, which significantly outperforms other SOTA methods.

It is also important to acknowledge certain limitations in our work. While we have established sufficient conditions for consistency, necessary conditions remain unaddressed, and the tightness of convergence rates for generalization error and excess risk is yet to be fully evidenced. The algorithm derived from our theory only represents a specific instance, leaving room for the exploration and comparison of diverse algorithms developed within this framework.

## Acknowledgments

This work was supported by the National Natural Science Foundation of China under Grant U24A20322, and Hong Kong UGC under grant UGC/FDS11/E02/22, UGC/FDS11/E03/24 and in part by the NSFC Excellent Young Scientists Fund 62422118. This research work was also supported by the Big Data Computing Center of Southeast University.

## Impact Statement

This paper presents work whose goal is to advance the field of ensemble clustering. There are many potential societal consequences of our work, none which we feel must be specifically highlighted here.

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

# Appendix

## A. Overview of the Appendix

Our appendix consists of three main sections:

- Proofs of the three theorems in Section 3, i.e., Theorem 3.1 (generalization error bound), Theorem 3.2 (excess risk bound), and Theorem 3.3 (sufficient conditions for consistency).

- Proof of Theorem 4.1 in Section 4, Eq. (7) $\Rightarrow$ Eq. (8) and Eq. (9) $\Rightarrow$ Eq. (12)

- Some information omitted from the main text due to the space limit, including pseudo code for Section 5.3, related work, details of datasets and comparison methods, clustering performance on ARI and Purity metrics, as well as experiments on hyper-parameters, ablation studies, and ensemble size analysis.

To clarify our proof process, we provide sketches of the proofs for the theorems in Sections 3 and 4 in Appendices A.1 and A.2, respectively, and the detailed proofs are provided in Appendix B.

### A.1. The sketching proof of Theorem 3.1, 3.2, 3.3

A.1.1. THE SKETCHING PROOF OF THEOREM 3.1

To proof Theorem 3.1, we first make the following decomposition,

$$\hat{F}\left(\hat{Z}; \bar{K}\right) - F\left(\mathcal{Z}; K^*\right) = \underbrace{\hat{F}\left(\hat{Z}; \bar{K}\right) - \hat{F}\left(\hat{\mathcal{Z}}; K^*\right)}_{\mathcal{A}} + \underbrace{\hat{F}\left(\hat{\mathcal{Z}}; K^*\right) - F\left(\mathcal{Z}; K^*\right)}_{\mathcal{B}},$$

where $\hat{F}(\hat{\mathcal{Z}}; K^*)$ is the empirical error with expected CA matrix $\mathbf{K}^*$ (function $K^*$),

$$\hat{F}(\hat{\mathcal{Z}}; K^*) = \max_{\{\hat{\zeta}_q\}_{q=1}^k \in \Gamma} \frac{1}{n^2} \sum_{q=1}^{k} \sum_{i=1}^{n} \sum_{j=1}^{n} K^*(x_i, x_j)\hat{\zeta}_q(x_i)\hat{\zeta}_q(x_j).$$

$\mathcal{A}$ can be further decomposed as (note that $\hat{F}\left(\hat{Z}; \bar{K}\right)$ is equivalent to $\hat{F}\left(\hat{\mathbf{Z}}; \bar{\mathbf{K}}\right)$, $\hat{F}\left(\hat{\mathcal{Z}}; K^*\right)$ is equivalent to $\hat{F}\left(\hat{\boldsymbol{\mathcal{Z}}}; \mathbf{K}^*\right)$)

$$
\begin{aligned}
&\hat{F}\left(\hat{Z}; \bar{K}\right) - \hat{F}\left(\hat{\mathcal{Z}}; K^*\right) \\
=& \frac{1}{n} \left( \operatorname{tr}\left(\hat{\mathbf{Z}}^\top \bar{\mathbf{K}} \hat{\mathbf{Z}}\right) - \operatorname{tr}\left(\hat{\boldsymbol{\mathcal{Z}}}^\top \mathbf{K}^* \hat{\boldsymbol{\mathcal{Z}}}\right) \right) \\
=& \underbrace{\frac{1}{n} \left( \operatorname{tr}\left(\hat{\mathbf{Z}}^\top \left(\bar{\mathbf{K}} - \mathbf{K}^*\right) \hat{\mathbf{Z}}\right) \right)}_{\mathcal{A}_1} + \underbrace{\frac{1}{n} \left( \operatorname{tr}\left(\hat{\mathbf{Z}}^\top \mathbf{K}^* \hat{\mathbf{Z}}\right) - \operatorname{tr}\left(\hat{\boldsymbol{\mathcal{Z}}}^\top \mathbf{K}^* \hat{\boldsymbol{\mathcal{Z}}}\right) \right)}_{\mathcal{A}_2}.
\end{aligned}
$$

Therefore, we prove Theorem 3.1 by bounding $\mathcal{A}_1, \mathcal{A}_2, \mathcal{B}$ separately, which leads to the following three lemmas.

**Lemma A.1.** *Under the general assumptions, we have*

$$\mathcal{A}_1 = \frac{1}{n} \left( \operatorname{tr}\left(\hat{\mathbf{Z}}^\top \left(\bar{\mathbf{K}} - \mathbf{K}^*\right) \hat{\mathbf{Z}}\right) \right) \leq \frac{2}{3m} \log \frac{2n}{\delta} + \sqrt{\frac{8}{m} \log \frac{2n}{\delta}},$$

*with probability at least $1 - \delta$. (The detailed proof of Lemma A.1 is in Appendix B.1)*

**Lemma A.2.** *Under the general assumptions and assume that the gap between the $k$-th and $(k+1)$-th eigenvalues of the expectation of normalized similarity matrix $\mathbf{K}^*$ is $\delta_k$ and $\delta_k \geq \frac{1}{c} > 0$ where $c$ is a constant, we have*

$$\mathcal{A}_2 = \frac{1}{n} \left( \operatorname{tr}\left(\hat{\mathbf{Z}}^\top \mathbf{K}^* \hat{\mathbf{Z}}\right) - \operatorname{tr}\left(\hat{\boldsymbol{\mathcal{Z}}}^\top \mathbf{K}^* \hat{\boldsymbol{\mathcal{Z}}}\right) \right) \leq 2\sqrt{2}c \left( \frac{2}{3m} \log \frac{2n}{\delta} + \sqrt{\frac{8}{m} \log \frac{2n}{\delta}} \right),$$

*with probability at least $1 - \delta$. (The detailed proof of Lemma A.2 is in Appendix B.2)*

**Lemma A.3.** *Under the general assumptions, we have*

$$\mathcal{B} = \hat{F}\left(\hat{\mathcal{Z}}; K^*\right) - F\left(\mathcal{Z}; K^*\right) \leq \frac{2\sqrt{2}\log(\frac{2}{\delta})}{\sqrt{n}},$$

*with probability at least $1 - \delta$. (The detailed proof of Lemma A.3 is in Appendix B.3)*

For Lemma A.1, our proof primarily relies on matrix Bernstein inequality. In the case of Lemma A.2, we apply perturbation theory to derive the bound for $\mathcal{A}_2$. The proof of Lemma A.3 is mainly concerned with the integral operator theory of (Rosasco et al., 2010). By combining Lemmas A.1, A.2, and A.3, we have

$$F\left(\hat{Z}; K^*\right) - F\left(\mathcal{Z}; K^*\right) \leq \left(2\sqrt{2}c + 1\right)\left(\frac{2}{3m}\log\frac{6n}{\delta} + \sqrt{\frac{8}{m}\log\frac{6n}{\delta}}\right) + \frac{2\sqrt{2}\log\left(\frac{6}{\delta}\right)}{\sqrt{n}}$$

with at least probability $1 - \delta$, which completes the proof of Theorem 3.1. $\qquad\square$

### A.1.2. THE SKETCHING PROOF OF THEOREM 3.2

Based on the proof of Theorem 3.1, we have

$$
\begin{aligned}
&F\left(\hat{Z}; K^*\right) - F\left(\mathcal{Z}; K^*\right)\\
=&F\left(\hat{Z}; K^*\right) - \hat{F}\left(\hat{Z}; \bar{K}\right) + \underbrace{\hat{F}\left(\hat{Z}; \bar{K}\right) - F\left(\mathcal{Z}; K^*\right)}_{\text{Generalization error}}\\
=&F\left(\hat{Z}; K^*\right) - \hat{F}\left(\hat{Z}; K^*\right) + \hat{F}\left(\hat{Z}; K^*\right) - \hat{F}\left(\hat{Z}; \bar{K}\right) + \underbrace{\hat{F}\left(\hat{Z}; \bar{K}\right) - \hat{F}\left(\hat{\mathcal{Z}}; K^*\right)}_{\mathcal{A}} + \underbrace{\hat{F}\left(\hat{\mathcal{Z}}; K^*\right) - F\left(\mathcal{Z}; K^*\right)}_{\mathcal{B}}\\
=&\underbrace{F\left(\hat{Z}; K^*\right) - \hat{F}\left(\hat{Z}; K^*\right)}_{\mathcal{C}} + \underbrace{\hat{F}\left(\hat{Z}; K^*\right) - \hat{F}\left(\hat{Z}; \bar{K}\right)}_{-\mathcal{A}_1} + \underbrace{\hat{F}\left(\hat{Z}; \bar{K}\right) - \hat{F}\left(\hat{Z}; K^*\right)}_{\mathcal{A}_1}\\
&+ \underbrace{\hat{F}\left(\hat{Z}; K^*\right) - \hat{F}\left(\hat{\mathcal{Z}}; K^*\right)}_{\mathcal{A}_2} + \underbrace{\hat{F}\left(\hat{\mathcal{Z}}; K^*\right) - F\left(\mathcal{Z}; K^*\right)}_{\mathcal{B}}\\
=&\underbrace{F\left(\hat{Z}; K^*\right) - \hat{F}\left(\hat{Z}; K^*\right)}_{\mathcal{C}} + \underbrace{\hat{F}\left(\hat{Z}; K^*\right) - \hat{F}\left(\hat{\mathcal{Z}}; K^*\right)}_{\mathcal{A}_2} + \underbrace{\hat{F}\left(\hat{\mathcal{Z}}; K^*\right) - F\left(\mathcal{Z}; K^*\right)}_{\mathcal{B}}.
\end{aligned}
$$

Therefore, we only need to bound $\mathcal{C}$, as the bounds of $\mathcal{A}_2$ and $\mathcal{B}$ can be obtained directly from Lemmas A.2, A.3.

**Lemma A.4.** *Under the general assumptions and with the additional condition that $||\hat{z}_q||_\infty \leq c_0$ ($c_0 > 0$ is a constant), we have*

$$\mathcal{C} = F(\hat{Z}; K^*) - \hat{F}(\hat{Z}; K^*) \leq k\left(\frac{2\sqrt{2}c_0}{\sqrt{n}} + \sqrt{\frac{8\log\frac{1}{\delta}}{n}}\right),$$

*with probability at least $1 - \delta$. (The detailed proof of Lemma A.4 is in Appendix B.4)*

For bounding $\mathcal{C}$, we utilize the McDiarmid's inequality (McDiarmid, 1989) and Rademacher complexity. The former is a standard tool to bound the difference of the random variable and its expectation. The reason for utilizing the latter technology is that, $\hat{F}(\hat{Z}; K^*)$ is a pairwise function and some tools in the i.i.d. condition is not satisfied (Li et al., 2023). We derive the bound of $\mathcal{C}$ analogously to (Li et al., 2023). By combining Lemmas A.2, A.3 and A.4, we derive that

$$F(\hat{Z}; K^*) - F(\mathcal{Z}; K^*) \leq k\left(\frac{2\sqrt{2}c_0}{\sqrt{n}} + \sqrt{\frac{8\log\frac{3}{\delta}}{n}}\right) + 2\sqrt{2}c\left(\frac{2}{3m}\log\frac{6n}{\delta} + \sqrt{\frac{8}{m}\log\frac{6n}{\delta}}\right) + \frac{2\sqrt{2}\log\left(\frac{6}{\delta}\right)}{\sqrt{n}}.$$

with probability at least $1 - \delta$. This concludes the proof of Theorem 3.2. $\qquad\square$

A.1.3. THE SKETCHING PROOF OF THEOREM 3.3

Theorem 3.3 describes that empirical eigenvectors ($\hat{\mathbf{z}}_q$) of CA matrix ($\bar{\mathbf{K}}$) converge to the eigenfunctions ($\zeta_q$) of integral operator ($L_{K^*}$) in probability in the limit case. We introduce the intermediate vector $\hat{\mathbf{z}}_q$ ($\hat{\mathbf{z}}_q$ is the eigenvector of expected CA matrix $\mathbf{K}^*$) and proceed with the following decomposition:

$$\|a_q\hat{\mathbf{z}}_q - \zeta_q\|_\infty \leq \underbrace{\|a_q\hat{\mathbf{z}}_q - b_q\hat{\mathbf{z}}_q\|_\infty}_{\mathcal{M}} + \underbrace{\|b_q\hat{\mathbf{z}}_q - \zeta_q\|_\infty}_{\mathcal{N}}.$$

For $\mathcal{N}$, we know that there exist a sequence $(b_q)_q \in \{-1, 1\}$ such that $\|b_q\hat{\mathbf{z}}_q - \zeta_q\|_\infty \to 0$ as $n \to \infty$, which has been proved in the Theorem 15 by (Von Luxburg et al., 2008). We need only prove that $\mathcal{M}$ converges to 0 in probability.

**Lemma A.5.** *Under the same assumptions as Lemma A.2, there exists a sequence $(a_q)_q \in \{-1, 1\}$ such that*

$$\|a_q\hat{\mathbf{z}}_q - b_q\hat{\mathbf{z}}_q\|_\infty \to 0,$$

*in probability as $m, n \to \infty$ and $m \gg \log n$. (The detailed proof of Lemma A.5 is in Appendix B.5)*

For Lemma A.5, our proof technique primarily relies on perturbation theory and trigonometric functions transformations. By incorporating $\mathcal{M}$, we complete the proof of Theorem 3.3. $\qquad\square$

## A.2. The sketching proof of Theorem 4.1 and Eq. (7) $\Rightarrow$ Eq. (8)

A.2.1. THE SKETCHING PROOF OF THEOREM 4.1

Theorem 4.1 presents the bias-diversity decomposition for ensemble clustering. To prove this theorem, we introduce the following two lemmas.

**Lemma A.6.** *According to the definitions in Section 4, where $\mathbf{K^w} = \sum_{t=1}^{m} w_t\mathbf{K}^{(i)}$, $\mathbf{K}^*$ is the expectation of $\mathbf{K}^{(i)}$, and $w_t$ is the weight of $t$-th base clusterings. We have the following decomposition*

$$\|\mathbf{K^w} - \mathbf{K}^*\|_\mathrm{F}^2 = \frac{1}{m}\sum_{t=1}^{m}\|mw_t\mathbf{K}^{(t)} - \mathbf{K}^*\|_\mathrm{F}^2 - \frac{1}{m}\sum_{t=1}^{m}\|mw_t\mathbf{K}^{(t)} - \mathbf{K^w}\|_\mathrm{F}^2.$$

*The detailed proof of Lemma A.6 is in Appendix B.6.*

**Lemma A.7.** *Under the same assumptions as Lemma A.2, we derive that*

$$\hat{F}\left(\hat{\mathbf{Z}}; \mathbf{K^w}\right) - \hat{F}\left(\hat{\boldsymbol{z}}; \mathbf{K}^*\right) \leq \left(\frac{k}{n} + \frac{2\sqrt{2}}{\lambda_k\left(\mathbf{K}^*\right) - \lambda_{k+1}\left(\mathbf{K}^*\right)}\right)\|\mathbf{K^w} - \mathbf{K}^*\|_\mathrm{F}.$$

*The detailed proof of Lemma A.7 is in Appendix B.7.*

The proof of Lemma A.6 primarily relies on the properties of the matrix trace. The proof of Lemma A.7 employs tools similar to those used in Lemma A.2. By combining these two lemmas and setting $c' = k/n + 2\sqrt{2}/(\lambda_k(\mathbf{K}^*) - \lambda_{k+1})$, we can readily prove Theorem 4.1. $\qquad\square$

A.2.2. THE SKETCHING PROOF OF EQ. (7) $\Rightarrow$ EQ. (8)

It can be observed that the coefficient $c'$ in Eq. (7) is a constant greater than zero (given the number of samples $n$, the number of base clusterings $m$, and the number of clusters $k$). Therefore, Eq. (7) is entirely equivalent to

$$\min_{\mathbf{w}} \sum_{t=1}^{m}\|\tilde{w}_t\mathbf{K}^{(t)} - \mathbf{K}^*\|_\mathrm{F}^2 - \sum_{t=1}^{m}\|\tilde{w}_t\mathbf{K}^{(t)} - \mathbf{K^w}\|_\mathrm{F}^2.$$

Through equivalent transformation, we arrive at the following lemma.

**Lemma A.8.** *With the same definition of Lemma A.6, we have*

$$\min_{\mathbf{w}} \sum_{t=1}^{m}\|\tilde{w}_t\mathbf{K}^{(t)} - \mathbf{K}^*\|_F^2 - \sum_{t=1}^{m}\|\tilde{w}_t\mathbf{K}^{(t)} - \mathbf{K^w}\|_\mathrm{F}^2 \;\Leftrightarrow\; \min_{\mathbf{w}} \; -2\mathrm{tr}\left(\mathbf{K^w}\mathbf{K}^*\right) + \mathrm{tr}\left(\mathbf{K^w}\mathbf{K^w}\right).$$

*The detailed proof of Lemma A.8 is in Appendix B.8.*

Through Lemma A.8, we can easily derive Eq. (8) from Eq. (7) . $\qquad\square$

A.2.3. THE SKETCHING PROOF OF EQ. (9) $\Rightarrow$ EQ. (12)

Eq. (9) is defined as

$$\max_{\mathbf{Z} \in \mathbb{R}^{n \times k}} 2\mathrm{tr}\left(\mathbf{K}^* \mathbf{Z}\mathbf{Z}^\top\right) + \min_{\mathbf{w}} \max_{\mathbf{Z} \in \mathbb{R}^{n \times k}} \mathrm{tr}\left(\mathbf{K}^{\mathbf{w}} \mathbf{Z}\mathbf{Z}^\top\right)$$
$$\text{s.t.} \mathbf{Z}^\top \mathbf{Z} = \mathbf{I}, \mathbf{w}^\top 1 = 1, \mathbf{w} \geq 0.$$

In this optimization problem, the first term does not contain the optimization variable $\mathbf{w}$, so we can directly combine it with the second term to obtain (we omit the constraints for the sake of brevity)

$$\min_{\mathbf{w}} \max_{\mathbf{Z} \in \mathbb{R}^{n \times k}} \left(\mathrm{tr}\left(\mathbf{K}^{\mathbf{w}} \mathbf{Z}\mathbf{Z}^\top\right) + 2\mathrm{tr}\left(\mathbf{K}^* \mathbf{Z}\mathbf{Z}^\top\right)\right).$$

Based on the properties of the matrix trace, we can derive that

$$\min_{\mathbf{w}} \max_{\mathbf{Z} \in \mathbb{R}^{n \times k}} \mathrm{tr}\left((2\mathbf{K}^* + \mathbf{K}^{\mathbf{w}}) \mathbf{Z}\mathbf{Z}^\top\right).$$

By replacing $\mathbf{K}^*$ with $\tilde{\mathbf{K}}$ in Eq. (11), we finally obtain Eq. (12).

# B. Detailed Proof

In this section, we provide detailed proofs for each lemma presented in Appendices A.1 and A.2.

## B.1. Proof of Lemma A.1

To prove Lemma A.1, we need to introduce the following matrix Bernstein inequality (Vershynin, 2018).

**Lemma B.1.** *(Matrix Bernstein Inequality) Let* $\mathbf{X}^{(1)}, \cdots, \mathbf{X}^{(m)}$ *be 0-mean* $n \times n$ *symmetric independent matrices such that* $\|\mathbf{X}^{(t)}\| \leq C$ *($C$ is a constant) almost surely for all $t$. Then, $\forall \varepsilon > 0$, we have*

$$P\left(\|\frac{1}{m}\sum_{t=1}^m \mathbf{X}^{(t)}\|_2 \geq \varepsilon\right) \leq 2n \exp\left\{-\frac{m^2 \varepsilon^2}{2\left(\sigma^2 + \frac{m\varepsilon C}{3}\right)}\right\},$$

*where* $\sigma^2 = \|\sum_{i=1}^m \mathbb{E}\left[\mathbf{X}^{(t)2}\right]\|_2$.

*Proof.* For $\mathcal{A}_1$, we have

$$\begin{aligned}
\mathcal{A}_1 &= \frac{1}{n}\left(\mathrm{tr}\left(\hat{\mathbf{Z}}^\top \left(\bar{\mathbf{K}} - \mathbf{K}^*\right) \hat{\mathbf{Z}}\right)\right) \\
&\leq \frac{n}{n}\left\|\hat{\mathbf{Z}}^\top \left(\bar{\mathbf{K}} - \mathbf{K}^*\right) \hat{\mathbf{Z}}^\top\right\|_2 \\
&\leq \left\|\hat{\mathbf{Z}}\right\|_2^2 \left\|\bar{\mathbf{K}} - \mathbf{K}^*\right\|_2 = \left\|\bar{\mathbf{K}} - \mathbf{K}^*\right\|_2
\end{aligned}$$

Define $\mathbf{X}^{(t)} = \mathbf{K}^{(t)} - \mathbf{K}^*$, obviously we have $\mathbb{E}[\mathbf{X}^{(t)}] = \mathbb{E}[\mathbf{K}^{(t)}] - \mathbf{K}^* = 0$. For $\sigma^2$, we have

$$\begin{aligned}
\sigma^2 &= \left\|\sum_{i=1}^m \mathbb{E}\left[\mathbf{X}^{(t)2}\right]\right\|_2 \\
&= \left\|\sum_{i=1}^m \mathbb{E}\left[\left(\mathbf{K}^{(t)} - \mathbf{K}^*\right)^2\right]\right\|_2 \\
&= \left\|\sum_{i=1}^m \left(\mathbb{E}\left[\mathbf{K}^{(t)2}\right] - \mathbb{E}\left[\mathbf{K}^{(t)}\mathbf{K}^*\right] - \mathbb{E}\left[\mathbf{K}^*\mathbf{K}^{(t)}\right] + \mathbb{E}\left[\mathbf{K}^{*2}\right]\right)\right\|_2 \\
&= \left\|\sum_{i=1}^m \mathbb{E}\left[\mathbf{K}^{(t)2}\right] - m\mathbf{K}^{*2}\right\|_2 \leq m \sup_t \left\|\mathbb{E}\left[\mathbf{K}^{(t)2}\right] - \mathbf{K}^{*2}\right\|_2 \\
&\leq m \sup_t \left\|\mathbb{E}[\mathbf{K}^{(t)2}]\right\|_2 + \left\|\mathbf{K}^{*2}\right\|_2
\end{aligned}$$

Based on Jensen's inequality and $\mathbf{K}^{(t)} \preceq \mathbf{I}$, $\mathbf{K}^* \preceq \mathbf{I}$, we have

$$\left\| \mathbb{E}\left[\mathbf{K}^{(k)2}\right] \right\|_2 \leq \mathbb{E}\left\|\mathbf{K}^{(t)2}\right\|_2 \leq \mathbb{E}\left\|\mathbf{K}^{(t)}\right\|_2^2 \leq \|\mathbf{I}\|_2^2, \quad \left\|\mathbf{K}^{*2}\right\|_2 \leq \|\mathbf{K}^*\|_2^2 \leq \|\mathbf{I}\|_2^2,$$

for any $t$. Therefore, we can bound $\sigma^2$ by

$$\sigma^2 \leq m \sup_t \left\| \mathbb{E}\left[\mathbf{K}^{(t)2}\right] - \mathbf{K}^{*2} \right\| \leq m \left\| \mathbb{E}\left[\mathbf{K}^{(t)2}\right] \right\|_2 + m \left\|\mathbf{K}^{*2}\right\|_2 \leq 2m \|\mathbf{I}\|_2^2 = 2m.$$

With **Lemma** B.1, we have

$$\mathcal{A}_1 \leq \frac{2}{3m} \log \frac{2n}{\delta} + \sqrt{\frac{8}{m} \log \frac{2n}{\delta}},$$

with probability at least $1 - \delta$. $\qquad\square$

## B.2. Proof of Lemma A.2

We use a variant of Davis-Kahan theory (Yu et al., 2014) to bound $\mathcal{A}_2$.

**Lemma B.2.** *(Davis-Kahan theory) Assume $\mathbf{X}$ and $\mathbf{X}'$ are two $n \times n$ real symmetric matrices and their largest $d$ eigenvalues are $\lambda_1 \geq \lambda_2 \geq \cdots \geq \lambda_d$ and $\lambda_1' \geq \lambda_2' \geq \cdots \geq \lambda_d'$, the matrices $\mathbf{Z}$ and $\mathbf{Z}'$ are composed of corresponding eigenvectors $\mathbf{Z} = [\mathbf{z}_1, \cdots, \mathbf{z}_d]$ and $\mathbf{Z}' = [\mathbf{z}_1', \cdots, \mathbf{z}_d']$, we have*

$$\|\sin\Theta\|_{\mathrm{F}} \leq \frac{2\min\left(d^{1/2}\|\mathbf{X} - \mathbf{X}'\|_2, \|\mathbf{X} - \mathbf{X}'\|_{\mathrm{F}}\right)}{\lambda_d - \lambda_{d+1}},$$

*where $\Theta = (\theta_1 = \cos^{-1}\sigma_1, \cdots, \theta_d = \cos^{-1}\sigma_d)^\top$, $\theta_1, \cdots, \theta_d$ are the singular values of $\mathbf{Z}^\top\mathbf{Z}'$, $\sin(\Theta)$ is the $d \times d$ diagonal matrix with the elements $\sin(\theta)_{ii} = \sin(\theta_i)$.*

*Proof.* For $\mathcal{A}_2$, we have

$$\begin{aligned}
\mathcal{A}_2 &= \frac{1}{n}\left(\mathrm{tr}\left(\hat{\mathbf{Z}}^\top\mathbf{K}^*\hat{\mathbf{Z}}\right) - \mathrm{tr}\left(\hat{\boldsymbol{\mathcal{Z}}}^\top\mathbf{K}^*\hat{\boldsymbol{\mathcal{Z}}}\right)\right) \\
&= \frac{1}{n}\|\mathbf{K}^*\|_{\mathrm{F}}\left\|\hat{\mathbf{Z}}\hat{\mathbf{Z}}^\top - \hat{\boldsymbol{\mathcal{Z}}}\hat{\boldsymbol{\mathcal{Z}}}^\top\right\|_{\mathrm{F}} \\
&\leq \|\mathbf{K}^*\|_2\left\|\hat{\mathbf{Z}}\hat{\mathbf{Z}}^\top - \hat{\boldsymbol{\mathcal{Z}}}\hat{\boldsymbol{\mathcal{Z}}}^\top\right\|_{\mathrm{F}} \\
&= \left\|\hat{\mathbf{Z}}\hat{\mathbf{Z}}^\top - \hat{\boldsymbol{\mathcal{Z}}}\hat{\boldsymbol{\mathcal{Z}}}^\top\right\|_{\mathrm{F}} \\
&= \sqrt{2}\left\|\sin\left(\Theta(\hat{\mathbf{Z}}, \hat{\boldsymbol{\mathcal{Z}}})\right)\right\|_{\mathrm{F}} \\
&\leq \frac{2\sqrt{2}\|\bar{\mathbf{K}} - \mathbf{K}^*\|_2}{\lambda_k(\mathbf{K}^*) - \lambda_{k+1}(\mathbf{K}^*)} \\
&\leq 2\sqrt{2}c\left(\frac{2}{3m}\log\frac{2n}{\delta} + \sqrt{\frac{8}{m}\log\frac{2n}{\delta}}\right),
\end{aligned}$$

with probability at least $1 - \delta$, where $c = \lambda_k(\mathbf{K}^*) - \lambda_{k+1}(\mathbf{K}^*)$. $\qquad\square$

## B.3. Proof of Lemma A.3

We introduce two integral operator in (Rosasco et al., 2010) to prove Lemma A.3.

Assume $\mathcal{H}$ is the RKHS associate with kernel function $K(x, y)$, the empirical covariance operator $T_n : \mathcal{H} \to \mathcal{H}$ is defined as

$$T_n = \frac{1}{n}\sum_{i=1}^n \langle \cdot, K_{x_i}\rangle K_{x_i},$$

where $K_{x_i} = K(x_i, \cdot)$. The expected covariance operator $T_{\mathcal{H}} : \mathcal{H} \to \mathcal{H}$ is

$$T_{\mathcal{H}} = \int_{\mathcal{X}} \langle K_x, \cdot\rangle K_x \mathrm{d}\rho(x).$$

*Proof.* By the definition of $\hat{F}\left(\hat{\mathcal{Z}};K^*\right)$ and $F\left(\mathcal{Z};K^*\right)$, we have

$$\hat{F}\left(\hat{\mathcal{Z}};K^*\right) - F\left(\mathcal{Z};K^*\right) = \frac{1}{n^2}\sum_{q=1}^{k}\sum_{i=1}^{n}\sum_{j=1}^{n}K^*\left(x_i,x_j\right)\hat{\zeta}_q\left(x_i\right)\hat{\zeta}_q\left(x_j\right) - \sum_{q=1}^{k}\iint_{\mathcal{X}}K^*\left(x,y\right)\zeta_q\left(x\right)\zeta_q\left(y\right)\mathrm{d}\rho\left(x\right)\mathrm{d}\rho\left(y\right)$$

$$= \frac{1}{n}\sum_{q=1}^{k}\sum_{i=1}^{n}\hat{\ell}_q\hat{\zeta}_q\left(x_i\right)\hat{\zeta}_q\left(x_i\right) - \sum_{q=1}^{k}\int_{\mathcal{X}}\ell_q\zeta_q\left(x\right)\zeta_q\left(x\right)\mathrm{d}\rho\left(x\right)$$

$$= \sum_{q=1}^{k}\hat{\ell}_q\hat{\boldsymbol{\zeta}}_q^{\top}\hat{\boldsymbol{\zeta}}_q - \sum_{q=1}^{k}\ell_q\int_{\mathcal{X}}\zeta_q\left(x\right)\zeta_q\left(x\right)\mathrm{d}\rho\left(x\right)$$

$$= \sum_{q=1}^{k}\left(\hat{\ell}_q - \ell_q\right)$$

where $\{\hat{\ell}_q\}_{q=1}^{k}, \{\ell_q\}_{q=1}^{k}$ are the largest $k$ eigenvalues of integral operators $\hat{L}_{K^*}\hat{\zeta}_q(x), L_{K^*}\zeta_q(x)$, respectively.

According to (Rosasco et al., 2010), the eigenvalues of $\hat{L}_K^*$ and $T_n$ (with kernel function $K^*$) are the same up to 0, so do $L_{K^*}$ and $T_{\mathcal{H}}$ (with kernel function $K^*$). Therefore, we have

$$\hat{F}\left(\hat{\mathcal{Z}};K^*\right) - F\left(\mathcal{Z};K^*\right) = \sum_{q=1}^{k}\left(\hat{\ell}_q - \ell_q\right) \leq \left|\sum_{q=1}^{k}\hat{\sigma}_q - \sigma_q\right| \leq \left|\mathrm{tr}\left(T_n\right) - \mathrm{tr}\left(T_{\mathcal{H}}\right)\right| \leq \frac{2\sqrt{2}\log\left(\frac{2}{\delta}\right)}{\sqrt{n}},$$

with probability at least $1 - \delta$. $\qquad\square$

## B.4. Proof of Lemma A.4

To prove Lemma A.4, we first introduce the McDiarmid's inequality.

**Lemma B.3.** *McDiarmid's inequality. For $m$ random variables $X_i \in \mathcal{X}, i \in [m]$, assume $f : \mathcal{X}^m \to \mathbb{R}$ is the real function of $X_i$ and $\forall\, x_1, \cdots, x_m, x_i' \in \mathcal{X}$, we have*

$$\left|f\left(x_1, \cdots, x_i, \cdots, x_m\right) - f\left(x_1, \cdots, x_i', \cdots, x_m\right)\right| \leq c_i,$$

*then $\forall\, \epsilon > 0$, the following inequality holds.*

$$P\left(f\left(X_1, \cdots, X_m\right) - \mathbb{E}\left[f\left(X_1, \cdots, X_m\right)\right] \geq \epsilon\right) \leq \exp\left\{\frac{-2\epsilon^2}{\sum_{i=1}^{m}c_i^2}\right\}.$$

As mentioned in Lemma A.4, $\hat{F}(\hat{Z};K^*)$ is a pairwise function and some tools in the i.i.d. condition is not satisfied, therefore, we make the following definition.

**Definition B.4.** *(Rademacher complexity for $\hat{F}(\hat{Z};K^*)$) Let $\mathcal{H}$ is the function space of $\hat{z}$, the empirical Rademacher complexity of $\mathcal{L}$ is definied as*

$$\hat{R}_n\left(\mathcal{L}\right) = \mathbb{E}_\sigma\left[\sup_{\hat{z}\in\mathcal{L}}\left|\frac{2}{\lfloor\frac{n}{2}\rfloor}\sum_{i=1}^{\lfloor\frac{n}{2}\rfloor}\sigma_i K^*\left(x_i, x_{i+\lfloor\frac{n}{2}\rfloor}\right)\hat{z}\left(x_i\right)\hat{z}\left(x_{i+\lfloor\frac{n}{2}\rfloor}\right)\right|\right],$$

where $\{\sigma_i\}_{i=1}^{\lfloor\frac{n}{2}\rfloor}$ are the i.i.d. Rademacher variables taking values $1$ and $-1$ with equal probability independent of the sample $S_n$. $\lfloor\frac{n}{2}\rfloor$ means the greatest integer less than or equal to $\frac{n}{2}$. The Rademacher complexity is the expectation of $\hat{R}_n(\mathcal{L})$, $R(\mathcal{L}) = \mathbb{E}[\hat{R}_n(\mathcal{L})]$.

*Proof.* Based on the definition of $\hat{Z}$, $\mathcal{C}$ can be reformulated as

$$
\mathcal{C} = F\left(\hat{Z}; K^*\right) - \hat{F}\left(\hat{Z}; K^*\right) = \sum_{q=1}^{k} \iint_{\mathcal{X}} K^*\left(x, y\right) \hat{z}_q\left(x\right) \hat{z}_q\left(y\right) \mathrm{d}\rho\left(x\right) \mathrm{d}\rho\left(y\right) - \frac{1}{n^2} \sum_{q=1}^{k} \sum_{i=1}^{n} \sum_{j=1}^{n} K^*\left(x_i, y_i\right) \hat{z}_q\left(x_i\right) \hat{z}_q\left(y_i\right)
$$

$$
= \sum_{q=1}^{k} \left( \mathbb{E}\left[K^*\left(x, y\right) \hat{z}_q\left(x\right) \hat{z}_q\left(y\right)\right] - \hat{\mathbb{E}}\left[K^*\left(x, y\right) \hat{z}_q\left(x\right) \hat{z}_q\left(y\right)\right] \right)
$$

$$
\leq k \sup_{\hat{z}_q \in \mathcal{L}} \left( \mathbb{E}[K^*(x, y)\hat{z}_q(x)\hat{z}_q(y)] - \hat{\mathbb{E}}[K^*(x, y)\hat{z}_q(x)\hat{z}_q(y)] \right).
$$

Assume the i.i.d. sampled data are $S_n = \{x_1, \cdots, x_i, \cdots, x_n\}$ and $S_n^{i,x_i'} = \{x_1, \cdots, x_i', \cdots, x_n\}$, we have

$$
\left| \sup_{\hat{z}_q \in \mathcal{L}} \left( \mathbb{E}[K^*(x, y)\hat{z}_q(x)\hat{z}_q(y)] - \hat{\mathbb{E}}_{S_n}[K^*(x, y)\hat{z}_q(x)\hat{z}_q(y)] \right) - \sup_{\hat{z}_q \in \mathcal{L}} \left( \mathbb{E}[K^*(x, y)\hat{z}_q(x)\hat{z}_q(y)] - \hat{\mathbb{E}}_{S_n^{i,x_i'}}[K^*(x, y)\hat{z}_q(x)\hat{z}_q(y)] \right) \right|
$$

$$
\leq \sup_{\hat{z}_q \in \mathcal{L}} \left| \hat{\mathbb{E}}_{S_n}\left[K^*\left(x, y\right) \hat{z}_q\left(x\right) \hat{z}_q\left(y\right)\right] - \hat{\mathbb{E}}_{S_n^{i,x_i'}}\left[K^*\left(x, y\right) \hat{z}_q\left(x\right) \hat{z}_q\left(y\right)\right] \right|
$$

$$
\leq \frac{2}{n^2} \sup_{\hat{z}_q \in \mathcal{L}} \sum_{j=1}^{n} \left( \left|K^*\left(x_i, x_j\right) \hat{z}_q\left(x_i\right) \hat{z}_q\left(x_j\right)\right| + \left|K^*\left(x_i', x_j\right) \hat{z}_q\left(x_i'\right) \hat{z}_q\left(x_j\right)\right| \right)
$$

$$
\leq \frac{2}{n^2} \sup_{\hat{z}_q \in \mathcal{L}} \sum_{j=1}^{n} \left( \left|\hat{z}_q\left(x_i\right) \hat{z}_q\left(x_j\right) + \hat{z}_q\left(x_i'\right) \hat{z}_q\left(x_j\right)\right| \right)
$$

$$
\leq \frac{4}{n}.
$$

The first inequality arises because $\sup_x \left(f\left(x\right) - g\left(x\right)\right) - \sup_x \left(f\left(x\right) - h\left(x\right)\right) \leq \sup_x \left(h\left(x\right) - g\left(x\right)\right)$; the second inequality is readily derived from $\left|f\left(x\right) - g\left(x\right)\right| \leq \left|f\left(x\right)\right| + \left|g\left(x\right)\right|$; concerning the third and fourth inequalities, we note that $K^*(x, y) \leq 1$ and $\sum_{j=1}^{n} \hat{z}_q\left(x_j\right) \hat{z}_q\left(x_j\right) = n$. Therefore, by applying McDiarmid's inequality, we have

$$
\sup_{\hat{z}_q \in \mathcal{L}} \left( \mathbb{E}\left[K^*\left(x, y\right) \hat{z}_q\left(x\right) \hat{z}_q\left(y\right)\right] - \hat{\mathbb{E}}_{S_n}\left[K^*\left(x, y\right) \hat{z}_q\left(x\right) \hat{z}_q\left(y\right)\right] \right)
$$

$$
\leq \mathbb{E}\left[ \sup_{\hat{z}_q \in \mathcal{L}} \left( \mathbb{E}\left[K^*\left(x, y\right) \hat{z}_q\left(x\right) \hat{z}_q\left(y\right)\right] - \hat{\mathbb{E}}_{S_n}\left[K^*\left(x, y\right) \hat{z}_q\left(x\right) \hat{z}_q\left(y\right)\right] \right) \right] + \sqrt{\frac{8 \log \frac{1}{\delta}}{n}},
$$

with probability at least $1-\delta$. Then we need to bound $\mathbb{E}\left[ \sup_{\hat{z}_q \in \mathcal{L}} \left( \mathbb{E}\left[K^*\left(x, y\right) \hat{z}_q\left(x\right) \hat{z}_q\left(y\right)\right] - \hat{\mathbb{E}}_{S_n}\left[K^*\left(x, y\right) \hat{z}_q\left(x\right) \hat{z}_q\left(y\right)\right] \right) \right]$.

According to (Clémençon et al., 2008), we have

$$
\mathbb{E}\left[ \sup_{\hat{z}_q \in \mathcal{L}} \left( \mathbb{E}\left[K^*\left(x, y\right) \hat{z}_q\left(x\right) \hat{z}_q\left(y\right)\right] - \hat{\mathbb{E}}_{S_n}\left[K^*\left(x, y\right) \hat{z}_q\left(x\right) \hat{z}_q\left(y\right)\right] \right) \right]
$$

$$
\leq \mathbb{E}\left[ \sup_{\hat{z}_q \in \mathcal{L}} \left( \mathbb{E}\left[K^*\left(x, y\right) \hat{z}_q\left(x\right) \hat{z}_q\left(y\right)\right] - \frac{1}{\lfloor \frac{n}{2} \rfloor} \sum_{i=1}^{\lfloor \frac{n}{2} \rfloor} K^*\left(x_i, x_{\lfloor \frac{n}{2} \rfloor + i}\right) \hat{z}_q\left(x_i\right) \hat{z}_q\left(x_{\lfloor \frac{n}{2} \rfloor + i}\right) \right) \right]
$$

Donate $S'_n = \{x'_1, \cdots, x'_n\}$ be the sampled i.i.d. data, $S'_n$ is independent of $S_n$. We have

$$\mathbb{E}\left[\sup_{\hat{z}_q \in \mathcal{L}} \left(\mathbb{E}\left[K^*(x,y)\,\hat{z}_q(x)\,\hat{z}_q(y)\right] - \frac{1}{\lfloor\frac{n}{2}\rfloor}\sum_{i=1}^{\lfloor\frac{n}{2}\rfloor} K^*\left(x_i, x_{\lfloor\frac{n}{2}\rfloor+i}\right)\hat{z}_q(x_i)\,\hat{z}_q\left(x_{\lfloor\frac{n}{2}\rfloor+i}\right)\right)\right]$$

$$=\mathbb{E}_{S_n}\left[\sup_{\hat{z}_q \in \mathcal{L}} \left(\mathbb{E}_{S'_n}\left[\frac{1}{\lfloor\frac{n}{2}\rfloor}\sum_{i=1}^{\lfloor\frac{n}{2}\rfloor} K^*\left(x'_i, x'_{\lfloor\frac{n}{2}\rfloor+i}\right)\hat{z}_q(x'_i)\,\hat{z}_q(x'_{\lfloor\frac{n}{2}\rfloor+i}) - \frac{1}{\lfloor\frac{n}{2}\rfloor}\sum_{i=1}^{\lfloor\frac{n}{2}\rfloor} K^*(x_i, x_{\lfloor\frac{n}{2}\rfloor+i})\hat{z}_q(x_i)\,\hat{z}_q\left(x_{\lfloor\frac{n}{2}\rfloor+i}\right)\right]\right)\right]$$

$$\leq\mathbb{E}_{S_n, S'_n}\left[\sup_{\hat{z}_q \in \mathcal{L}} \left(\frac{1}{\lfloor\frac{n}{2}\rfloor}\sum_{i=1}^{\lfloor\frac{n}{2}\rfloor}\left(K^*\left(x'_i, x'_{\lfloor\frac{n}{2}\rfloor+i}\right)\hat{z}_q(x'_i)\,\hat{z}_q\left(x'_{\lfloor\frac{n}{2}\rfloor+i}\right) - K^*\left(x_i, x_{\lfloor\frac{n}{2}\rfloor+i}\right)\hat{z}_q(x_i)\,\hat{z}_q\left(x_{\lfloor\frac{n}{2}\rfloor+i}\right)\right)\right)\right]$$

$$=\mathbb{E}_{S_n, S'_n, \sigma}\left[\sup_{\hat{z}_q \in \mathcal{L}} \left(\frac{1}{\lfloor\frac{n}{2}\rfloor}\sum_{i=1}^{\lfloor\frac{n}{2}\rfloor}\sigma_i\left(K^*\left(x'_i, x'_{\lfloor\frac{n}{2}\rfloor+i}\right)\hat{z}_q(x'_i)\,\hat{z}_q\left(x'_{\lfloor\frac{n}{2}\rfloor+i}\right) - K^*\left(x_i, x_{\lfloor\frac{n}{2}\rfloor+i}\right)\hat{z}_q(x_i)\,\hat{z}_q\left(x_{\lfloor\frac{n}{2}\rfloor+i}\right)\right)\right)\right]$$

$$=\frac{2}{\lfloor\frac{n}{2}\rfloor}\mathbb{E}_{S'_n, \sigma}\left[\sup_{\hat{z}_q \in \mathcal{L}} \left(\sum_{i=1}^{\lfloor\frac{n}{2}\rfloor}\sigma_i K^*\left(x'_i, x'_{\lfloor\frac{n}{2}\rfloor+i}\right)\hat{z}_q(x'_i)\,\hat{z}_q\left(x'_{\lfloor\frac{n}{2}\rfloor+i}\right)\right)\right]$$

$$\leq\frac{2}{\lfloor\frac{n}{2}\rfloor}\mathbb{E}_{S'_n}\left[\left(\sup_{\hat{z}_q \in \mathcal{L}}\sum_{i=1}^{\lfloor\frac{n}{2}\rfloor}\left(K^*\left(x'_i, x'_{\lfloor\frac{n}{2}\rfloor+i}\right)\hat{z}_q(x'_i)\,\hat{z}_q\left(x'_{\lfloor\frac{n}{2}\rfloor+i}\right)\right)^2\right)^{\frac{1}{2}}\right],$$

where $\{\sigma_i\}_{i=1}^{\lfloor\frac{n}{2}\rfloor}$ are the Rademacher variables. The second inequality is derived from Jensen's inequality; the third equality uses the standard symmetrization technique and the last inequality utilizes the Khinchin-Kahane inequality (Latała & Oleszkiewicz, 1994). Assume that $||\hat{z}_q||_\infty < \sqrt{c_0}$, we can obtain that

$$\frac{2}{\lfloor\frac{n}{2}\rfloor}\mathbb{E}_{S'_n}\left[\left(\sup_{\hat{z}_q \in \mathcal{L}}\sum_{i=1}^{\lfloor\frac{n}{2}\rfloor}\left(K^*\left(x'_i, x'_{\lfloor\frac{n}{2}\rfloor+i}\right)\hat{z}_q(x'_i)\,\hat{z}_q\left(x'_{\lfloor\frac{n}{2}\rfloor+i}\right)\right)^2\right)^{\frac{1}{2}}\right],$$

$$\leq\frac{2}{\lfloor\frac{n}{2}\rfloor}C\sqrt{\lfloor\frac{n}{2}\rfloor}$$

$$\leq\frac{2\sqrt{2}c_0}{\sqrt{n}}.$$

Thus, we can obtain

$$\mathcal{C} = F(\hat{Z}; K^*) - \hat{F}(\hat{Z}; K^*) \leq k\left(\frac{2\sqrt{2}c_0}{\sqrt{n}} + \sqrt{\frac{8\log\frac{1}{\delta}}{n}}\right).$$

with at least probability $1 - \delta$. $\qquad\square$

### B.5. Proof of Lemma A.5

*Proof.* For a given sequence $(a_q)_q$, $a_q\hat{\mathbf{z}}_q$ and $b_q\hat{\mathbf{z}}_q$, we can always find another sequence $(b_q)_q$ such that $\cos\theta(a_q\hat{\mathbf{z}}_q, b_q\hat{\mathbf{z}}_q) \geq 0$. Therefore, without loss of generality, we assume that the angle between $\hat{\mathbf{z}}_q$ and $\hat{z}_q$ is within $[0, \frac{\pi}{2}]$.

$$\|a_q\hat{\mathbf{z}}_q - b_q\hat{\mathbf{z}}_q\|_\infty \leq \|a_q\hat{\mathbf{z}}_q - b_q\hat{\mathbf{z}}_q\|_2 = \sqrt{2 - 2\cos\theta} = \sqrt{4\sin^2\left(\frac{\theta}{2}\right)} = 2\left|\sin\left(\frac{\theta}{2}\right)\right|.$$

With $\sin(\theta) = 2\sin\left(\frac{\theta}{2}\right)\cos\left(\frac{\theta}{2}\right)$ and $\cos\left(\frac{\theta}{2}\right) \geq \frac{1}{\sqrt{2}}$, we have

$$\|a_q\hat{\mathbf{z}}_q - b_q\hat{\mathbf{z}}_q\|_\infty \leq \sqrt{2}\sin(\theta),$$

where $\theta = \theta(a_q\hat{\mathbf{z}}_q, b_q\hat{\mathbf{z}}_q)$. From the proof of Lemma A.2, we can readily deduce that as $m, n \to \infty, m \gg \log n$, $\|a_q\hat{\mathbf{z}}_q - b_q\hat{\mathbf{z}}_q\|_\infty \to 0$. $\qquad\square$

## B.6. Proof of Lemma A.6

*Proof.* For $\|\mathbf{K}^{\mathbf{w}} - \mathbf{K}^*\|_{\mathrm{F}}^2$, we have

$$
\begin{aligned}
&\|\mathbf{K}^{\mathbf{w}} - \mathbf{K}^*\|_{\mathrm{F}}^2 \\
=& 2\|\mathbf{K}^* - \mathbf{K}^{\mathbf{w}}\|_{\mathrm{F}}^2 - \|\mathbf{K}^* - \mathbf{K}^{\mathbf{w}}\|_{\mathrm{F}}^2 \\
=& 2\mathrm{tr}\left((\mathbf{K}^* - \mathbf{K}^{\mathbf{w}})^\top \left(\mathbf{K}^* - \frac{1}{m}\sum_{t=1}^m mw_t\mathbf{K}^{(t)}\right)\right) - \|\mathbf{K}^* - \mathbf{K}^{\mathbf{w}}\|_{\mathrm{F}}^2 \\
=& \frac{1}{m}\sum_{t=1}^m 2\mathrm{tr}\left((\mathbf{K}^* - \mathbf{K}^{\mathbf{w}})^\top \left(\mathbf{K}^* - mw_t\mathbf{K}^{(t)}\right)\right) - \|\mathbf{K}^* - \mathbf{K}^{\mathbf{w}}\|_{\mathrm{F}}^2 \\
=& \frac{1}{m}\sum_{t=1}^m \left(-2\mathrm{tr}\left((\mathbf{K}^* - \mathbf{K}^{\mathbf{w}})^\top \left(mw_t\mathbf{K}^{(t)} - \mathbf{K}^*\right)\right) - \|\mathbf{K}^* - \mathbf{K}^{\mathbf{w}}\|_{\mathrm{F}}^2\right) \\
=& \frac{1}{m}\sum_{t=1}^m \left(-2\mathrm{tr}\left((\mathbf{K}^* - \mathbf{K}^{\mathbf{w}})^\top \left(mw_t\mathbf{K}^{(t)} - \mathbf{K}^*\right)\right) - \|\mathbf{K}^* - \mathbf{K}^{\mathbf{w}}\|_{\mathrm{F}}^2\right) \\
& + \frac{1}{m}\sum_{t=1}^m \left(-\|mw_t\mathbf{K}^{(t)} - \mathbf{K}^*\|_{\mathrm{F}}^2 + \|mw_t\mathbf{K}^{(t)} - \mathbf{K}^*\|_{\mathrm{F}}^2\right) \\
=& \frac{1}{m}\sum_{t=1}^m \left(-\|\mathbf{K}^* - \mathbf{K}^{\mathbf{w}} + mw_t\mathbf{K}^{(t)} - \mathbf{K}^*\|_{\mathrm{F}}^2 + \|mw_t\mathbf{K}^{(t)} - \mathbf{K}^*\|_{\mathrm{F}}^2\right) \\
=& \frac{1}{m}\sum_{t=1}^m \|mw_t\mathbf{K}^{(t)} - \mathbf{K}^*\|_{\mathrm{F}}^2 - \frac{1}{m}\sum_{t=1}^m \|mw_t\mathbf{K}^{(t)} - \mathbf{K}^{\mathbf{w}}\|_{\mathrm{F}}^2.
\end{aligned}
$$

This concludes the proof of Lemma A.6. $\qquad\square$

## B.7. Proof of Lemma A.7

*Proof.* Based on Lemma B.2, we have

$$
\begin{aligned}
&\hat{F}\left(\hat{\mathbf{Z}}; \mathbf{K}^{\mathbf{w}}\right) - \hat{F}\left(\hat{\boldsymbol{\mathcal{Z}}}; \mathbf{K}^*\right) \\
=& \frac{1}{n}\mathrm{tr}\left(\hat{\mathbf{Z}}^\top \mathbf{K}^{\mathbf{w}} \hat{\mathbf{Z}}\right) - \frac{1}{n}\mathrm{tr}\left(\hat{\boldsymbol{\mathcal{Z}}}^\top \mathbf{K}^* \hat{\boldsymbol{\mathcal{Z}}}\right) \\
=& \frac{1}{n}\left(\mathrm{tr}\left(\hat{\mathbf{Z}}^\top \left(\mathbf{K}^{\mathbf{w}} - \mathbf{K}^*\right)\hat{\mathbf{Z}}\right)\right) + \frac{1}{n}\mathrm{tr}\left(\mathbf{K}^*\left(\hat{\mathbf{Z}}\hat{\mathbf{Z}}^\top - \hat{\boldsymbol{\mathcal{Z}}}\hat{\boldsymbol{\mathcal{Z}}}^\top\right)\right) \\
\leq& \frac{1}{n}\|\mathbf{K}^{\mathbf{w}} - \mathbf{K}^*\|_{\mathrm{F}}\|\hat{\mathbf{Z}}\|_{\mathrm{F}}^2 + \frac{1}{n}\|\mathbf{K}^*\|_{\mathrm{F}}\|\hat{\mathbf{Z}}\hat{\mathbf{Z}}^\top - \hat{\boldsymbol{\mathcal{Z}}}\hat{\boldsymbol{\mathcal{Z}}}^\top\|_{\mathrm{F}} \\
\leq& \frac{k}{n}\|\mathbf{K}^{\mathbf{w}} - \mathbf{K}^*\|_{\mathrm{F}} + \frac{n}{n}\|\hat{\mathbf{Z}}\hat{\mathbf{Z}}^\top - \hat{\boldsymbol{\mathcal{Z}}}\hat{\boldsymbol{\mathcal{Z}}}^\top\|_{\mathrm{F}} \\
\leq& \frac{k}{n}\|\mathbf{K}^{\mathbf{w}} - \mathbf{K}^*\|_{\mathrm{F}} + \frac{2\sqrt{2}}{\lambda_k\left(\mathbf{K}^*\right) - \lambda_{k+1}\left(\mathbf{K}^*\right)}\|\mathbf{K}^{\mathbf{w}} - \mathbf{K}^*\|_2 \\
\leq& \left(\frac{k}{n} + \frac{2\sqrt{2}}{\lambda_k\left(\mathbf{K}^*\right) - \lambda_{k+1}\left(\mathbf{K}^*\right)}\right)\|\mathbf{K}^{\mathbf{w}} - \mathbf{K}^*\|_{\mathrm{F}}
\end{aligned}
$$

The first inequality utilizes the properties of the matrix trace, and the second inequality holds because $\hat{\mathbf{Z}}$ is an $n \times k$ column-orthogonal matrix, and $\|\mathbf{K}^*\|_{\mathrm{F}} \leq n\|\mathbf{K}^*\|_2 \leq n$ (noting that $\mathbf{K}^*$ is a degree normalized matrix). This concludes the proof of Lemma A.7. $\qquad\square$

## B.8. Proof of Lemma A.8

*Proof.* Note that $\tilde{w} = mw_t$ and $\mathbf{K}^{\mathbf{w}} = \sum_{t=1}^{m} w_t \mathbf{K}^{(t)}$, we have

$$
\min_{\mathbf{w}} \ \sum_{t=1}^{m} \|\tilde{w}_t \mathbf{K}^{(t)} - \mathbf{K}^*\|_{\mathrm{F}}^2 - \sum_{t=1}^{m} \|\tilde{w}_t \mathbf{K}^{(t)} - \mathbf{K}^{\mathbf{w}}\|_{\mathrm{F}}^2
$$

$$
\Leftrightarrow \min_{\mathbf{w}} \ \sum_{t=1}^{m} \left( \|mw_t \mathbf{K}^{(t)}\|_{\mathrm{F}}^2 - 2\mathrm{tr}\left(mw_t \mathbf{K}^{(t)} \mathbf{K}^*\right) + \|\mathbf{K}^*\|_{\mathrm{F}}^2 \right) - \sum_{t=1}^{m} \left( \|mw_t \mathbf{K}^{(t)}\|_{\mathrm{F}}^2 - 2\mathrm{tr}\left(mw_t \mathbf{K}^{(t)} \mathbf{K}^{\mathbf{w}}\right) + \|\mathbf{K}^{\mathbf{w}}\|_{\mathrm{F}}^2 \right)
$$

$$
\Leftrightarrow \min_{\mathbf{w}} \ \sum_{t=1}^{m} \left( -2\mathrm{tr}\left(mw_t \mathbf{K}^{(t)} \mathbf{K}^*\right) + \|\mathbf{K}^*\|_{\mathrm{F}}^2 \right) - \sum_{t=1}^{m} \left( -2\mathrm{tr}\left(mw_t \mathbf{K}^{(t)} \mathbf{K}^{\mathbf{w}}\right) + \|\mathbf{K}^{\mathbf{w}}\|_{\mathrm{F}}^2 \right)
$$

$$
\Leftrightarrow \min_{\mathbf{w}} \ -2m\mathrm{tr}\left(\mathbf{K}^{\mathbf{w}} \mathbf{K}^*\right) - \sum_{t=1}^{m} \left( -2\mathrm{tr}\left(mw_t \mathbf{K}^{(t)} \mathbf{K}^{\mathbf{w}}\right) + \|\mathbf{K}^{\mathbf{w}}\|_{\mathrm{F}}^2 \right)
$$

$$
\Leftrightarrow \min_{\mathbf{w}} \ -2m\mathrm{tr}\left(\mathbf{K}^{\mathbf{w}} \mathbf{K}^*\right) + 2m\mathrm{tr}\left(\mathbf{K}^{\mathbf{w}} \mathbf{K}^{\mathbf{w}}\right) - m\mathrm{tr}\left(\mathbf{K}^{\mathbf{w}} \mathbf{K}^{\mathbf{w}}\right)
$$

$$
\Leftrightarrow \min_{\mathbf{w}} \ -2\mathrm{tr}\left(\mathbf{K}^{\mathbf{w}} \mathbf{K}^*\right) + \mathrm{tr}\left(\mathbf{K}^{\mathbf{w}} \mathbf{K}^{\mathbf{w}}\right)
$$

Note that in the proof, since ignoring $\mathbf{K}^*$ does not affect the optimization of $\mathbf{w}$, we have omitted term $\|\mathbf{K}^*\|_{\mathrm{F}}$. This concludes the proof of Lemma A.8. $\qquad\square$

## C. Pseudo code for Section 5.3

---

**Algorithm 1**

---

**Input:** Base clusterings $\{\mathbf{K}^{(t)}\}_{t=1}^{m}$.
**Initialization:** Weight $\{w_t\}_t^m = \frac{1}{m}$, the number of cluster $k$, hyper-parameter $\alpha$, the number of iterations $p$.
**Output:** Clustering result.

1: **while** not converged **do**
2:   Compute the matrix $\mathbf{Z}^{(p)}$ (consists of the eigenvectors corresponding to the top $k$ largest eigenvalues of $\mathbf{K}^{\mathbf{w}^{(p)}}$).
3:   Calculate $\frac{\partial \mathcal{J}(\mathbf{w}^{(p)})}{\partial w_t}$ and the descent direction $d_t^{(p)}$ in Eq. (13).
4:   Update $\mathbf{w}^{(p+1)}$ as $\mathbf{w}^{(p+1)} \leftarrow \mathbf{w}^{(p)} + \beta \mathbf{d}^{(p)}$.
5:   **if** $|\mathbf{w}^{(p+1)} - \mathbf{w}^{(p)}| \leq \epsilon$ **then**
6:     Break.
7:   **end if**
8:   $p \leftarrow p + 1$.
9: **end while**
10: Apply the $k$-means algorithm to $\mathbf{Z}^{(p)}$ to obtain the final clustering result.

---

## D. Related Work

This section reviews related works on generalization performance of ensemble clustering. In (Liu et al., 2017), the author derived the generalization error bound of ensemble clustering with finite base clusterings from the perspective of weighted kernel $k$-means. Denote $\mathbf{B}_{n \times (\sum_{t=1}^{m} k^{(t)})}$ as a combined binary matrix of $m$ base clusterings where

$$\mathbf{B}(x, \cdot) = b(x) = < b(x)_1, \cdots, b(x)_m >,$$
$$b(x)_t = < b(x)_{t1}, \cdots, b(x)_{tk^{(t)}} >,$$
$$b(x)_{ti} \begin{cases} 1, & \text{if} \quad \pi^{(t)}(x) = i \\ 0, & \text{else} \end{cases}.$$

Based on the above definition, the author derived ensemble clustering is equivalent to weighted kernel $k$-means algorithm,

$$\hat{F}\left(\hat{\mathbf{Z}}\right) = \max_{\hat{\mathbf{Z}}} \frac{1}{k} \text{tr}\left(\mathbf{Z}^\top \mathbf{D}^{-1/2} \mathbf{S} \mathbf{D}^{-1/2} \mathbf{Z}\right) \Leftrightarrow \hat{G}(x) = \sum_{x \in \mathcal{X}} g_{m_1, \cdots, m_k}(x),$$

where $\mathbf{S}$ is the CA matrix, $g_{m_1, \cdots, m_k}(x) = \min_k ||\frac{b(x)}{w_b(x)} - m_k||^2$, $m_k = \frac{\sum_{x \in C_k} b(x)}{\sum_{x \in C_k} w_{b(x)}}$, and $w_b(x) = \mathbf{D}(x, x) = \sum_{t=1}^{m} \sum_{i=1}^{n} \delta(\pi^{(t)}(x), \pi^{(t)}(x_i))$. The generalization error bound of ensemble clustering with finite base clusterings is

$$\mathbb{E}_x g_{m_1, \cdots, m_k}(x) - \frac{1}{n} \sum_{i=1}^{n} g_{m_1, \cdots, m_k}(x_i)$$

$$\leq \frac{\sqrt{2\pi} mk}{n} \left(\sum_{i=1}^{n} \left(w_{b(x_i)}\right)^{-2}\right)^{\frac{1}{2}} + \frac{\sqrt{8\pi} mk}{\sqrt{n} \min_{x \in \mathcal{X}} w_{b(x)}} + \frac{\sqrt{2\pi} mk}{n \min_{x \in \mathcal{X}} \left(w_{b(x)}\right)^2} \left(\sum_{i=1}^{n} \left(w_{b(x_i)}\right)^2\right)^{\frac{1}{2}} + \left(\frac{\ln(1/\delta)}{2n}\right)^{\frac{1}{2}},$$

with probability $1 - \delta$. To the best of our knowledge, this work is the only one that provides a generalization error bound for ensemble clustering. Other theoretical analyses related to clustering include the generalization performance of multi-view clustering. For example, (Liu, 2023) proposed SimpleMKKM algorithm in multi-view clustering and derived its generalization error. (Liang et al., 2023) demonstrated the consistency of kernel weights in multi-view clustering and derived its the excess risk bound. Nevertheless, the scenarios they consider are remain limited to finite ensembles.

## E. Comparative Experiment

In this section, we provide additional details about the comparative experiments that are omitted in the main text due to space limitations.

## E.1. Details of Datasets

In the comparative experiments in Section 6.1, we used 10 benchmark datasets including images, DNA, sensor information, etc. We have summarized the feature information of the datasets in Table 3, and the detailed information is as follows:

1. **Phishing Websites**[1]: The dataset consists of a collection of legitimate and phishing website instances. Each website is represented by the set of features which denote, whether the website is legitimate or not.

2. **Rice**[2]: A total of 3810 images of rice grains were captured from two species: Cammeo and Osmancik rices. For each grain in these images, seven morphological features were extracted.

3. **TOX_171**[3]: The dataset contains 171 samples, each with 5748 features, derived from feature selection at Arizona State University's repository.

4. **Obesity**[4]: The dataset contains 2111 instances from individuals in the countries of Mexico, Peru, and Colombia. It includes 16 features reflecting eating habits and physical conditions, designed to estimate obesity levels.

5. **Seeds**[5]: The dataset includes measurements of the geometrical properties of kernels from three wheat varieties, with seven real-valued features extracted using a soft X-ray technique and the GRAINS package.

6. **ALLAML**[6]: The dataset consists of a DNA microarray data matrix, where rows represent genes and columns represent cancer patients diagnosed with one of two types of leukemia: AML or ALL. The elements of the matrix indicate gene expression levels in the corresponding patients.

7. **warpAR10P**[7]: The dataset includes over 4000 color images of 126 individuals, comprising 70 men and 56 women. It captures various facial expressions, lighting conditions, and occlusions.

8. **WFRN**[8]: The dataset includes four sensor readings, termed "simplified distances" (*i.e.* front, left, right and back). Each distance represents the minimum sensor reading within a 60-degree arc in the respective direction around the robot.

9. **Abalone**[9]: The dataset is designed to predict the age of abalones by collecting eight physical measurements, including sex, length, diameter, height, whole weight, shucked weight, viscera weight and shell weight.

10. **Website Phishing**[10]: The dataset includes 1353 websites, with phishing URLs sourced from the Phishtank data archive and legitimate websites collected from Yahoo and starting point directories using a custom PHP web script. It comprises 548 legitimate websites, 702 phishing URLs and 103 suspicious URLs.

## E.2. Details Method

The detailed descriptions of 9 comparison methods introduced in Section 6.1 are as follows.

1. CEAM (Zhou et al., 2024), this method introduces a novel approach for clustering ensemble which refines weak base clustering results through diffusion on an adaptive multiplex structure.

2. CEs$^2$L, CEs$^2$Q (Li et al., 2019), these two methods use a linear determinacy function and a quadratic determinacy function to assess sample stability in clustering ensemble respectively, distinguishing stable samples (cluster core) from less stable ones (cluster halo) for robust clustering.

---

[1] http://archive.ics.uci.edu/dataset/327/phishing+websites
[2] http://archive.ics.uci.edu/dataset/545/rice+cammeo+and+osmancik
[3] https://github.com/jundongl/scikit-feature/blob/master/skfeature/data/TOX-171.mat
[4] https://archive.ics.uci.edu/dataset/544/estimation+of+obesity+levels+based+on+eating+habits+and+physical+condition
[5] https://archive.ics.uci.edu/dataset/236/seeds
[6] https://github.com/jundongl/scikit-feature/blob/master/skfeature/data/ALLAML.mat
[7] https://github.com/jundongl/scikit-feature/blob/master/skfeature/data/warpAR10P.mat
[8] https://archive.ics.uci.edu/dataset/194/wall+following+robot+navigation+data
[9] https://archive.ics.uci.edu/dataset/1/abalone
[10] https://archive.ics.uci.edu/dataset/379/website+phishing

*Table 3.* Size of different datasets

| No. | Dataset | #Instance | #Feature | #Class |
|---|---|---|---|---|
| D1 | Phishing Websites | 2456 | 30 | 2 |
| D2 | Rice | 3810 | 7 | 2 |
| D3 | TOX_171 | 171 | 5748 | 4 |
| D4 | Obesity | 2111 | 16 | 7 |
| D5 | Seeds | 210 | 7 | 3 |
| D6 | ALLAML | 72 | 7129 | 2 |
| D7 | warpAR10P | 130 | 2400 | 10 |
| D8 | WFRN | 5456 | 4 | 4 |
| D9 | Abalone | 4177 | 8 | 3 |
| D10 | Website Phishing | 1353 | 9 | 3 |

*Table 4.* Performance (%) evaluation of different datasets based on the ARI metric. We have highlighted the values of the best-performing method in **bold**, and the second-best method is marked with an underline.

| Method | D1 | D2 | D3 | D4 | D5 | D6 | D7 | D8 | D9 | D10 | Average |
|---|---|---|---|---|---|---|---|---|---|---|---|
| CEAM (TKDE'24) | $6.6_{\pm 12}$ | $42.8_{\pm 31}$ | $12.9_{\pm 4}$ | $20.4_{\pm 1}$ | $59.0_{\pm 13}$ | $2.7_{\pm 5}$ | $2.5_{\pm 1}$ | $10.8_{\pm 4}$ | $12.8_{\pm 5}$ | $10.1_{\pm 7}$ | $18.1_{\pm 8}$ |
| CEs$^2$L (AIJ'19) | $2.4_{\pm 4}$ | $3.0_{\pm 10}$ | $14.0_{\pm 3}$ | $20.3_{\pm 2}$ | $33.3_{\pm 19}$ | $18.3_{\pm 6}$ | $0.2_{\pm 2}$ | $6.8_{\pm 7}$ | $15.4_{\pm 4}$ | $9.6_{\pm 9}$ | $12.3_{\pm 7}$ |
| CEs$^2$Q (AIJ'19) | $1.7_{\pm 3}$ | $3.5_{\pm 7}$ | $12.4_{\pm 3}$ | $20.0_{\pm 2}$ | $31.2_{\pm 17}$ | $18.5_{\pm 6}$ | $0.3_{\pm 2}$ | $9.0_{\pm 4}$ | $15.2_{\pm 3}$ | $6.7_{\pm 5}$ | $11.8_{\pm 6}$ |
| LWEA (TCYB'18) | $-0.5_{\pm 0}$ | $62.9_{\pm 4}$ | $13.1_{\pm 3}$ | $21.2_{\pm 1}$ | $57.5_{\pm 5}$ | $18.5_{\pm 6}$ | $0.0_{\pm 2}$ | $10.0_{\pm 4}$ | $13.5_{\pm 3}$ | $8.8_{\pm 6}$ | $20.5_{\pm 4}$ |
| NWCA (arXiv'24) | $-0.5_{\pm 0}$ | $62.3_{\pm 4}$ | $12.9_{\pm 2}$ | $21.6_{\pm 1}$ | $56.3_{\pm 6}$ | $19.8_{\pm 5}$ | $-0.1_{\pm 2}$ | $10.4_{\pm 1}$ | $13.3_{\pm 3}$ | $11.7_{\pm 6}$ | $20.8_{\pm 3}$ |
| ECCMS (TNNLS'24) | $-0.5_{\pm 0}$ | $56.1_{\pm 24}$ | $13.5_{\pm 3}$ | $21.3_{\pm 1}$ | $60.8_{\pm 7}$ | $19.0_{\pm 6}$ | $-0.3_{\pm 1}$ | $\underline{12.2}_{\pm 4}$ | $14.0_{\pm 3}$ | $10.5_{\pm 6}$ | $20.7_{\pm 6}$ |
| MKKM (arXiv'18) | $8.8_{\pm 14}$ | $47.1_{\pm 25}$ | $9.5_{\pm 2}$ | $14.2_{\pm 5}$ | $53.8_{\pm 10}$ | $13.6_{\pm 12}$ | $2.1_{\pm 2}$ | $7.2_{\pm 3}$ | $10.9_{\pm 6}$ | $10.1_{\pm 7}$ | $17.7_{\pm 8}$ |
| SMKKM (TPAMI'23) | $8.8_{\pm 5}$ | $41.9_{\pm 10}$ | $14.6_{\pm 3}$ | $17.0_{\pm 3}$ | $55.5_{\pm 11}$ | $13.2_{\pm 9}$ | $\underline{3.5}_{\pm 1}$ | $7.2_{\pm 4}$ | $\underline{15.7}_{\pm 2}$ | $12.2_{\pm 5}$ | $19.0_{\pm 5}$ |
| SEC (TKDE'17) | $8.9_{\pm 15}$ | $23.8_{\pm 25}$ | $12.8_{\pm 4}$ | $13.5_{\pm 5}$ | $26.9_{\pm 19}$ | $13.5_{\pm 12}$ | $1.1_{\pm 2}$ | $5.6_{\pm 7}$ | $7.2_{\pm 6}$ | $5.2_{\pm 5}$ | $11.9_{\pm 9}$ |
| Fix $\alpha = 0.1$ | $\underline{30.9}_{\pm 15}$ | $\underline{69.2}_{\pm 1}$ | $\underline{15.8}_{\pm 4}$ | $\underline{22.1}_{\pm 2}$ | $\underline{67.5}_{\pm 5}$ | $\underline{20.6}_{\pm 5}$ | $2.6_{\pm 1}$ | $12.0_{\pm 5}$ | $14.8_{\pm 5}$ | $\underline{14.5}_{\pm 6}$ | $\underline{27.0}_{\pm 4}$ |
| Proposed | $\mathbf{30.9}_{\pm 15}$ | $\mathbf{69.2}_{\pm 1}$ | $\mathbf{16.7}_{\pm 3}$ | $\mathbf{22.1}_{\pm 2}$ | $\mathbf{67.5}_{\pm 5}$ | $\mathbf{21.5}_{\pm 5}$ | $\mathbf{4.2}_{\pm 1}$ | $\mathbf{18.6}_{\pm 2}$ | $\mathbf{16.0}_{\pm 3}$ | $\mathbf{14.5}_{\pm 6}$ | $\mathbf{28.1}_{\pm 3}$ |

*Table 5.* Performance (%) evaluation of different datasets based on the F-score metric. We have highlighted the values of the best-performing method in **bold**, and the second-best method is marked with an underline.

| Method | D1 | D2 | D3 | D4 | D5 | D6 | D7 | D8 | D9 | D10 | Average |
|---|---|---|---|---|---|---|---|---|---|---|---|
| CEAM (TKDE'24) | $60.5_{\pm 9}$ | $79.4_{\pm 15}$ | $46.4_{\pm 3}$ | $42.7_{\pm 1}$ | $83.1_{\pm 7}$ | $66.0_{\pm 2}$ | $22.8_{\pm 2}$ | $51.4_{\pm 3}$ | $49.9_{\pm 3}$ | $61.8_{\pm 6}$ | $56.4_{\pm 5}$ |
| CEs$^2$L (AIJ'19) | $58.0_{\pm 4}$ | $59.5_{\pm 7}$ | $46.3_{\pm 2}$ | $42.0_{\pm 2}$ | $65.0_{\pm 12}$ | $72.2_{\pm 3}$ | $19.3_{\pm 2}$ | $49.1_{\pm 5}$ | $51.7_{\pm 3}$ | $62.7_{\pm 6}$ | $52.6_{\pm 4}$ |
| CEs$^2$Q (AIJ'19) | $57.4_{\pm 3}$ | $60.3_{\pm 5}$ | $44.7_{\pm 3}$ | $41.9_{\pm 2}$ | $62.9_{\pm 12}$ | $72.4_{\pm 3}$ | $19.2_{\pm 1}$ | $50.5_{\pm 5}$ | $51.6_{\pm 3}$ | $60.3_{\pm 4}$ | $52.1_{\pm 4}$ |
| LWEA (TCYB'18) | $55.5_{\pm 0}$ | $89.6_{\pm 1}$ | $46.0_{\pm 3}$ | $43.2_{\pm 1}$ | $81.7_{\pm 4}$ | $72.4_{\pm 3}$ | $18.6_{\pm 2}$ | $49.5_{\pm 1}$ | $51.3_{\pm 2}$ | $61.2_{\pm 4}$ | $56.9_{\pm 2}$ |
| NWCA (arXiv'24) | $55.5_{\pm 0}$ | $89.4_{\pm 1}$ | $45.9_{\pm 2}$ | $43.6_{\pm 1}$ | $80.7_{\pm 5}$ | $73.2_{\pm 2}$ | $18.8_{\pm 2}$ | $49.2_{\pm 1}$ | $51.2_{\pm 2}$ | $63.5_{\pm 4}$ | $57.1_{\pm 2}$ |
| ECCMS (TNNLS'24) | $55.5_{\pm 0}$ | $85.6_{\pm 12}$ | $46.1_{\pm 3}$ | $43.3_{\pm 1}$ | $84.0_{\pm 3}$ | $72.6_{\pm 3}$ | $18.5_{\pm 2}$ | $51.0_{\pm 3}$ | $51.6_{\pm 3}$ | $62.5_{\pm 4}$ | $57.1_{\pm 3}$ |
| MKKM (arXiv'18) | $62.1_{\pm 10}$ | $82.6_{\pm 11}$ | $42.9_{\pm 3}$ | $37.4_{\pm 5}$ | $79.8_{\pm 7}$ | $70.8_{\pm 5}$ | $\underline{25.2}_{\pm 3}$ | $50.2_{\pm 2}$ | $49.7_{\pm 6}$ | $62.5_{\pm 6}$ | $56.3_{\pm 6}$ |
| SMKKM (TPAMI'23) | $62.9_{\pm 4}$ | $73.7_{\pm 7}$ | $47.7_{\pm 3}$ | $39.8_{\pm 2}$ | $80.6_{\pm 8}$ | $69.9_{\pm 4}$ | $23.4_{\pm 3}$ | $53.2_{\pm 1}$ | $\underline{52.2}_{\pm 1}$ | $63.3_{\pm 4}$ | $56.7_{\pm 4}$ |
| SEC (TKDE'17) | $62.2_{\pm 10}$ | $71.9_{\pm 12}$ | $46.0_{\pm 3}$ | $37.2_{\pm 4}$ | $59.9_{\pm 13}$ | $71.0_{\pm 4}$ | $20.5_{\pm 2}$ | $48.2_{\pm 5}$ | $45.7_{\pm 5}$ | $58.8_{\pm 5}$ | $52.1_{\pm 6}$ |
| Fix $\alpha = 0.1$ | $\underline{76.5}_{\pm 9}$ | $\underline{91.6}_{\pm 0}$ | $\underline{48.9}_{\pm 3}$ | $\underline{43.7}_{\pm 1}$ | $\underline{87.6}_{\pm 2}$ | $\underline{73.8}_{\pm 3}$ | $21.4_{\pm 2}$ | $\underline{55.4}_{\pm 5}$ | $51.5_{\pm 4}$ | $\underline{65.1}_{\pm 5}$ | $\underline{61.5}_{\pm 3}$ |
| Proposed | $\mathbf{76.5}_{\pm 9}$ | $\mathbf{91.6}_{\pm 1}$ | $\mathbf{50.0}_{\pm 2}$ | $\mathbf{43.7}_{\pm 1}$ | $\mathbf{87.6}_{\pm 2}$ | $\mathbf{73.8}_{\pm 2}$ | $\mathbf{27.4}_{\pm 3}$ | $\mathbf{63.3}_{\pm 1}$ | $\mathbf{52.3}_{\pm 2}$ | $\mathbf{65.1}_{\pm 5}$ | $\mathbf{63.1}_{\pm 3}$ |

3. LWEA (Huang et al., 2018), this method enhances ensemble clustering by employing a local weighting strategy based on cluster uncertainty and an ensemble-driven validity measure.

4. NWCA (Zhang et al., 2024), this method discovers that smaller clusters have higher precision and proposes the normalized ensemble entropy to weight different clusters accordingly.

5. ECCMS (Jia et al., 2024), this method enhances co-association matrices in ensemble clustering by extracting high-confidence pairings from base clusterings and propagating them to refine the CA matrix.

6. MKKM (Bang et al., 2018), this method utilizes a min-max model to manage adversarial perturbations, ensuring the identification of accurate clusterings by optimally balancing the influence of multiple data views.

7. SMKKM (Liu, 2023), this method transforms a complex min-max problem into a simpler minimization of an optimal value function, optimizing kernel coefficients and clustering matrices effectively to achieve robust clustering performance.

8. SEC (Liu et al., 2017), this method combines the strengths of the co-association matrix with the efficiency of weighted K-means clustering and derives its generalization error bound.

### E.3. Details of Comparative Experiment

In the appendix, we continue to demonstrate the performance of the algorithm on ARI and Purity. As can be seen in Table 4 and Table 5, on both ARI and Purity, our method consistently leads against the compared methods across all datasets. For example, on the D1 (Phishing) dataset, our method achieves an ARI of 30.9%, while the second-best method only reaches 8.9%; in terms of Purity, ours is at 76.5%, whereas the second-best is at 62.9%. Moreover, even with fixed hyper-parameter, our method outperforms others on these two metrics, and while it may not be the second-best method on some datasets, such as D8 (WFRN), it is only slightly weaker than the second-best method (with a 0.2% difference in ARI).

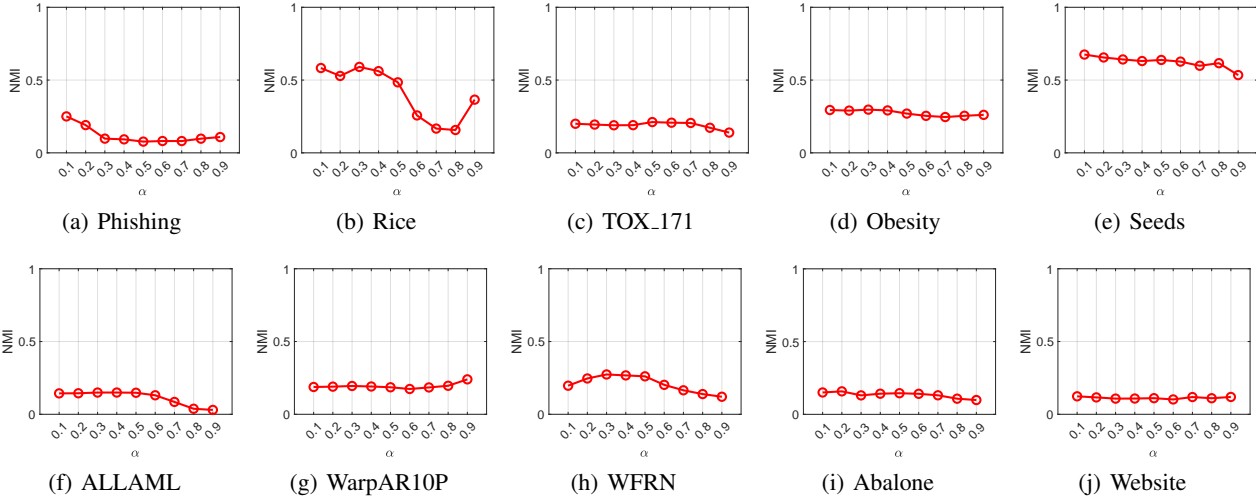

*Figure 4.* Analysis of hyperparameter $\alpha$ in $\tilde{\mathbf{K}}$. We vary the value of $\alpha$ from 0.1 to 0.9, with an incremental step of 0.1.

### E.4. Hyper-parameter Analysis

In this paper, we have only one hyper-parameter, $\alpha$, which serves as the threshold for extracting high-confidence elements. Fig. 4 shows the performance of our model under different $\alpha$ settings. It can be seen that our method is quite robust across most datasets, and the optimal hyper-parameter is generally between 0.1 and 0.3. From the comparative experiments, we can also see that even with fixed parameters, our algorithm performs well. Therefore, we think that our algorithm is robust to the hyper-parameter $\alpha$.

### E.5. Ablation Experiment

Table 6 presents the results of our ablation experiments. When the Bias term is removed, the model becomes completely dominated by Diversity, which may drive the optimization process away from the correct direction and cause significant performance degradation; when the Diversity term is removed, the model loses the ability to leverage differences among base clusterings, leading to noticeable performance drops. Overall, in the vast majority of datasets, removing either Bias or

*Table 6.* Ablation experiments (clustering performance: %). We separately remove the Bias term (denoted as w/o Bias) and the Diversity term (denoted as w/o Diversity) from the original model to observe changes in the model's performance across three metrics.

| Method | D1 | D2 | D3 | D4 | D5 | D6 | D7 | D8 | D9 | D10 |
|---|---|---|---|---|---|---|---|---|---|---|
| | | | | | NMI | | | | | |
| Proposed | **25.0**$_{\pm12}$ | **59.0**$_{\pm1}$ | **21.1**$_{\pm3}$ | **29.4**$_{\pm2}$ | **67.5**$_{\pm3}$ | **15.0**$_{\pm4}$ | 22.9$_{\pm2}$ | **27.5**$_{\pm2}$ | **15.8**$_{\pm3}$ | **12.4**$_{\pm4}$ |
| w/o Bias | 8.7$_{\pm4}$ | 38.5$_{\pm11}$ | 19.3$_{\pm4}$ | 27.0$_{\pm2}$ | 59.4$_{\pm9}$ | 10.5$_{\pm5}$ | 20.0$_{\pm2}$ | 18.2$_{\pm3}$ | 15.5$_{\pm2}$ | 10.5$_{\pm4}$ |
| w/o Diversity | 23.7$_{\pm11}$ | 40.8$_{\pm13}$ | 19.5$_{\pm3}$ | 29.0$_{\pm2}$ | 67.5$_{\pm4}$ | 14.7$_{\pm4}$ | **25.1**$_{\pm2}$ | 17.2$_{\pm1}$ | 7.3$_{\pm2}$ | 12.1$_{\pm4}$ |
| | | | | | ARI | | | | | |
| Proposed | **30.9**$_{\pm15}$ | **69.5**$_{\pm2}$ | **16.7**$_{\pm3}$ | **22.1**$_{\pm2}$ | **67.5**$_{\pm5}$ | **21.5**$_{\pm5}$ | **4.2**$_{\pm1}$ | **18.6**$_{\pm2}$ | **16.0**$_{\pm3}$ | **14.5**$_{\pm6}$ |
| w/o Bias | 8.8$_{\pm5}$ | 41.9$_{\pm10}$ | 14.6$_{\pm3}$ | 17.0$_{\pm3}$ | 55.5$_{\pm11}$ | 13.2$_{\pm9}$ | 3.5$_{\pm1}$ | 7.2$_{\pm4}$ | 15.7$_{\pm2}$ | 12.2$_{\pm5}$ |
| w/o Diversity | 6.9$_{\pm7}$ | 64.9$_{\pm12}$ | 15.0$_{\pm3}$ | 21.9$_{\pm3}$ | 58.0$_{\pm4}$ | 21.4$_{\pm5}$ | 3.1$_{\pm1}$ | 5.0$_{\pm1}$ | 5.6$_{\pm3}$ | 14.1$_{\pm6}$ |
| | | | | | Purity | | | | | |
| Proposed | **76.5**$_{\pm9}$ | **91.7**$_{\pm1}$ | **50.0**$_{\pm2}$ | **43.7**$_{\pm1}$ | **87.6**$_{\pm2}$ | **73.8**$_{\pm2}$ | 27.4$_{\pm3}$ | **63.3**$_{\pm1}$ | **52.3**$_{\pm2}$ | **65.1**$_{\pm5}$ |
| w/o Bias | 62.9$_{\pm4}$ | 73.7$_{\pm7}$ | 47.7$_{\pm3}$ | 39.8$_{\pm2}$ | 80.6$_{\pm8}$ | 69.9$_{\pm4}$ | 23.4$_{\pm3}$ | 53.2$_{\pm1}$ | 52.2$_{\pm1}$ | 63.3$_{\pm4}$ |
| w/o Diversity | 74.4$_{\pm11}$ | 84.2$_{\pm6}$ | 47.3$_{\pm2}$ | 43.1$_{\pm1}$ | 87.2$_{\pm3}$ | 73.8$_{\pm2}$ | **27.7**$_{\pm2}$ | 52.8$_{\pm1}$ | 45.0$_{\pm2}$ | 64.8$_{\pm5}$ |

Diversity results in inferior clustering performance, while the complete model achieves the best results, confirming that the joint optimization of both components is essential.

### E.6. Ensemble Size Analysis

Figure 5 reports the results of all methods across different datasets by varying the ensemble size $m$ in terms of NMI. It can be observed that our method outperforms the compared SOTA methods on almost all datasets, except for the ALLAML and Abalone datasets when the ensemble size $m$ is 10. Additionally, it is evident that the performance of our method generally improves as $m$ increases, which aligns with the conclusion derived from Theorem 3.1.

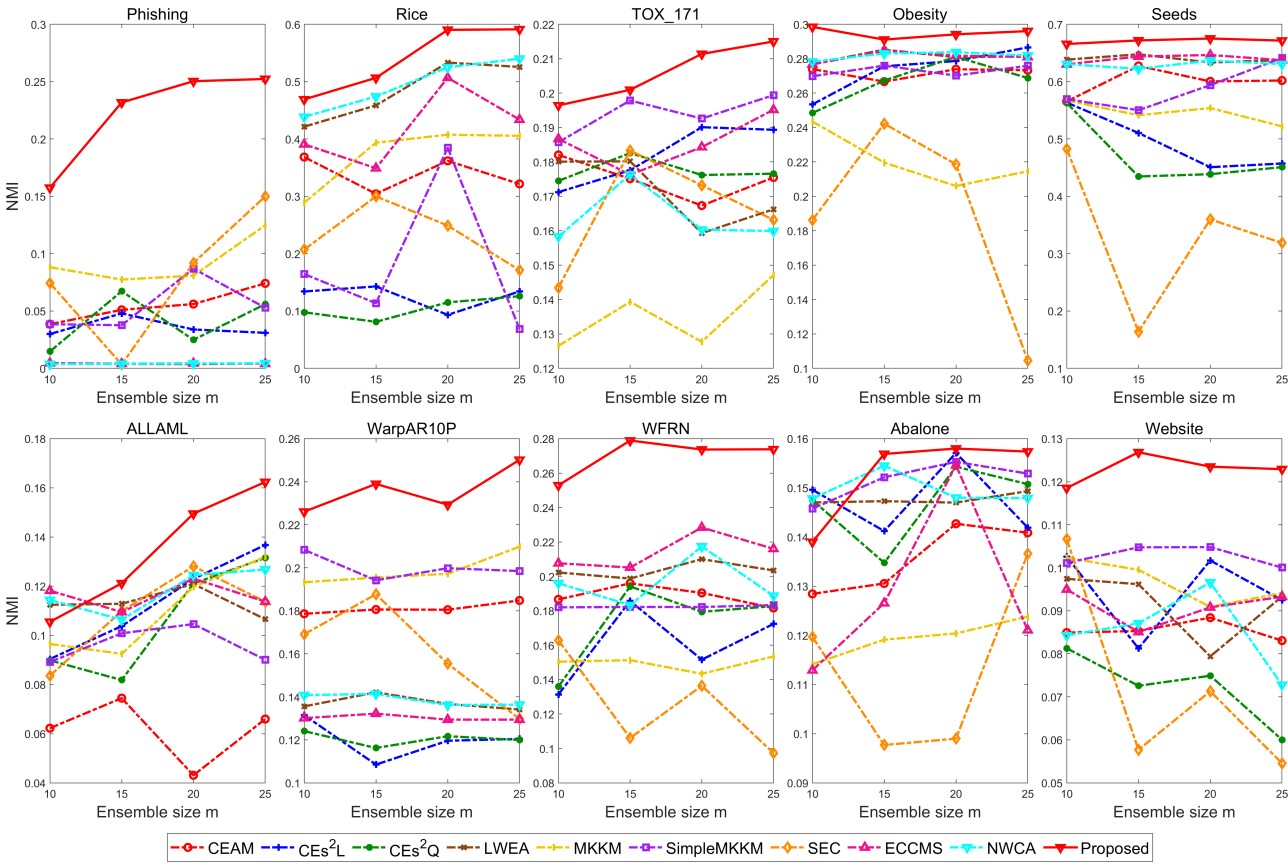

*Figure 5.* On each dataset, we vary the number of base clusterings $m$ in the ensemble and observe the corresponding changes in performance, as measured by NMI.

