# OpenReview forum: "Generalization Performance of Ensemble Clustering: From Theory to Algorithm"
_ICML.cc/2025/Conference — ICML 2025 poster_

### Official Review · Reviewer_smn4 · 2025-03-03

**Overall Recommendation:** 5

**Summary:**

This paper explores the theoretical foundations of ensemble clustering, focusing on its generalization performance, including generalization error, excess risk, and consistency. The authors derive convergence rates for both generalization error and excess risk, which are bounded by $\mathcal{O}(\sqrt{(\log n / m)} + 1/\sqrt{n})$ ($n,m$ are the numbers of samples and base clusterings) and demonstrate that ensemble clustering achieves consistency when both m and n approach infinity, and $m \gg \log n$. Recognizing that $m, n$ are finite in practice, the authors theoretically demonstrate that better clustering performance can be achieved by minimizing the bias of base clustering from its expectation and maximizing the diversity among base clusterings. Based on this, they instantiate their theory to a novel algorithm that utilized high-confidence pairwise similarity to approximate the expected clustering and solve it using a reduced gradient descent method, achieving state-of-the-art performance.
Key contributions include:
(1) For the first time, the authors derive the theoretical guarantees for generalization error, excess risk, and consistency of ensemble clustering;
(2) They develop a bias-diversity decomposition and innovatively establish the relationship between diversity and robustness in ensemble clustering;
(3) They propose a practical algorithm validated through extensive experiments. This work bridges theory and practice, offering both rigorous analysis and a high-performing solution for ensemble clustering.

**Claims And Evidence:**

The methods proposed in the paper are well-aligned with the ensemble clustering problem and their claims made in this paper are supported by both theoretical analysis and experiments. I think the theoretical results in this paper are clear, providing generalization error, excess risk bounds, and consistency. The exploration of Bias and Diversity in ensemble clustering is well-developed, with a solid theoretical foundation and sufficient experimental validation.

**Essential References Not Discussed:**

I have reviewed the references cited in the paper and did not find any significant omissions.

**Experimental Designs Or Analyses:**

The authors provide several important theoretical claims related to ensemble clustering, specifically regarding its generalization performance, excess risk, and consistency. Besides, they provide a bias-diversity decomposition for ensemble clustering under their designed objective function, along with a proof of the equivalence between diversity and robustness. I have reviewed all the proofs provided, and the theoretical claims in this paper appear to be solid.

**Methods And Evaluation Criteria:**

The methods proposed in the paper are well-aligned with the ensemble clustering problem. The authors define the objective function of ensemble clustering in the form of a spectrum, which is logical and reasonable. Using this objective function, the authors investigate its generalization performance, including generalization error, excess risk, and consistency. These theoretical insights are of significant importance for research in ensemble clustering. Unlike heuristic definitions of the objective function, the authors instantiate their theory to develop a new algorithm. I believe this is highly valuable and provides a fresh perspective for the theoretical study of ensemble clustering. The adopted benchmark datasets (10 real datasets) and evaluation criteria (NMI, ARI, Purity) are appropriate for measuring clustering performance and the experiments they designed are reasonable and useful.

**Other Comments Or Suggestions:**

I don’t have other comments or suggestions.

**Other Strengths And Weaknesses:**

Strengths
Originality: The paper offers a novel theoretical framework for ensemble clustering.
Quality: The theoretical results are rigorous and well-supported by comprehensive experiments.
Clarity: The paper is clearly written, with detailed explanations of complex concepts and a well-structured presentation of the theoretical derivations and experimental results.
Significance: The findings contribute valuable insights into the practical application of ensemble clustering, offering guidance for optimizing clustering performance in real-world scenarios.
Weaknesses
1.The paper would benefit from providing statistical significance tests to confirm the robustness of the reported improvements (e.g., paired t-tests or Wilcoxon signed-rank tests).
2.The details of some experiments are not fully detailed in the main text but are mentioned to be in the appendices. It would be beneficial to include a brief summary of key findings from these experiments in the main paper.
3.Although the authors have conducted extensive theoretical analysis of their algorithm, I believe it is necessary to add a part discussing the time complexity of their algorithm, as this would help to understand its practical implementation in real-world scenarios.
4.The authors' derivation from Equation (8) to Equation (9) seems somewhat abrupt. I would suggest they include more details to clarify the process.

**Questions For Authors:**

Besides the questions I mentioned in the Weakness section, I would like to ask the authors whether they have considered more general cases when instantiating their theory, beyond just this spectral form of loss function?

**Relation To Broader Scientific Literature:**

This paper significantly advances the theoretical understanding of ensemble clustering, offering theoretical guidance for practical applications. It establishes a formal link between model diversity and robustness, providing a theoretical foundation for enhancing performance in various ensemble-based methods. Notably, the algorithm introduced is not heuristic but directly derived from their theoretical framework, likely increasing scholarly focus on theoretical research.

**Theoretical Claims:**

The authors provide several important theoretical claims related to ensemble clustering, specifically regarding its generalization performance, excess risk, and consistency. Besides, they provide a bias-diversity decomposition for ensemble clustering under their designed objective function, along with a proof of the equivalence between diversity and robustness. I have reviewed all the proofs provided, and the theoretical claims in this paper appear to be solid.

---

> ### Author Rebuttal · Authors · 2025-04-01
>
> **We sincerely thanks for all your constructive comments!**
> >**Weakness 1. Statistical significance tests**
>
>
> We conducted paired t-tests on our method using NMI, ARI, and Purity metrics, and the results show that our method significantly **outperforms the sota methods** compared across almost all datasets. We believe this further demonstrates the effectiveness of the proposed approach. The experiment is in <https://anonymous.4open.science/r/ICML8598/TabB345.pdf>
> >**Weakness 2. Describe some key conclusions in the main text**
>
> Due to page space limitations, we placed some conclusions in the appendix. Following your suggestion, we will move the key conclusions from Appendix E.4. Hyper-parameter Analysis, E.5. Ablation Experiment, and E.6. Ensemble Size Analysis into the main text and noted that readers can find the detailed analyses in the appendix.
> >**Weakness 3: Weakness 2. Time complexity of the proposed algorithm**
>
> The time complexity of our method is composed of the following parts: Construct each similarity, Calculate $\tilde{K}$, Solve $Z$, Compute reduced gradient, and Update $w$. Through theoretical analysis, the **time complexity** of our method is $O(n^{2.376})$. Detailed time complexities for each module can be found in the comments for Reviewer 2. Additionally, the time complexities of most baseline methods we compared are also $O(n^{2.376})$, yet our method **significantly outperforms** them. We also conducted time cost experiments on different datasets, and the results confirm that our method surpasses these Co-associate matrix optimization-based methods in terms of both performance and time efficiency. The experiment can be found in <https://anonymous.4open.science/r/ICML8598/FigA4.pdf>
> >**Weakness 4: Clarify formula derivation (Eq. (8) to Eq. (9))**
>
> For $\min_w -2tr(K^wK^*)$, our objective is to adjust the weights ${w}$ such that the weighted CA matrix $K^w$ closely approximates its expected value $K^*$. To this end, we **reformulate the problem as finding a low-dimensional embedding** $Z$ for $K^*$. This approach consolidates 1) $max_w tr(K^wK^* )$ and 2) $\max_Z tr(K^* ZZ^T)$ into a single expression, $max_Z tr(K^*ZZ^T)$, which corresponds to the Bias term in the objective function of Eq. (9). For $\min_w tr(K^wK^w)$, we apply a similar strategy. Specifically, for the term $\min_w tr(K^wK^w)$, we replace one instance of $K^w$ with the low-dimensional embedding $Z$, i.e., $tr(K^wK^w)\Rightarrow
>  \max_Z tr(K^wZZ)$, and transforming it into a min-max optimization problem. Additionally, we constrain $Z$ to be an orthogonal matrix in its columns. It is important to note that the original constraint $w^Tw=1$ is nonconvex and a standard relaxation technique is $w^Tw\le 1$. However, we revise this to $w1 = 1$ and also modify the definition of $K^w$. This approach has the advantage of allowing $w$ to be better interpreted as a weight distribution.
> >**Question 1: Generalization form of loss function**
>
> There is a **general framework** for our method. For a continuously differentiable and strongly convex function $\phi$, let $\Omega$ be a convex set where $x, y \in \Omega\$, we can define the Bregman divergence as:
> $$D_\phi(x, y) = \phi(x) - \phi(y) - \langle \nabla \phi(y), (x-y) \rangle. $$
>
> Thus, we can transform Eq. (7) in the text into a more generalized form:
> $$D_{\phi}(K^*, K^w) = \frac{1}{m} \sum_{t=1}^m D_{\phi}(K^*, mw_tK^t) - \frac{1}{m} \sum_{t=1}^m D_{\phi}(K^w, mw_tK^t),$$
>
> where $K^w = (\nabla \phi)^{-1}\left( \frac{1}{m} \sum_{t=1}^m \nabla \phi(mw_tK^t) \right) $.
>
> Consequently, we can derive a generalized Bias-Diversity decomposition:
> $$\underset{w}{\min} \, -\langle \nabla \phi(K^w), K^* \rangle + \langle \nabla \phi(K^w), K^w \rangle - \phi(K^w)$$
> Based on this, we can let $\phi(x)$ be various metric functions, such as KL divergence, JS divergence, etc. We believe **this is a more significant conclusion** and will be further discussed in future work.

---

### Official Review · Reviewer_kz2j · 2025-03-07

**Overall Recommendation:** 3

**Summary:**

This paper investigates the theoretical foundations of ensemble clustering, focusing on its generalization performance, including generalization error, excess risk, and consistency. The authors derive theoretical bounds for these indicators and propose a new ensemble clustering algorithm based on their findings, demonstrating significant improvements over existing methods. The key contributions and findings are as follows:

1.	The paper establishes the convergence rate for generalization error and excess risk, showing that increasing the number of base clusterings helps reduce the generalization error but cannot eliminate it. Furthermore, it proves that when both the number of samples nnn and base clusterings mmm approach infinity, with m>>log n, ensemble clustering achieves uniform convergence, meaning the clustering result progressively approximates the true data structure.

2.	The study reveals that clustering performance can be improved by minimizing the bias of base clusterings (i.e., the difference between each base clustering and its expectation) while maximizing diversity among them. The authors further establish that maximizing diversity closely relates to robust optimization models.

3.	Leveraging this theoretical framework, the authors introduce a novel ensemble clustering algorithm. It utilizes high-confidence elements to approximate the expected co-association matrix and formulates clustering as a min-max optimization problem. The algorithm optimizes the base clustering weights using a descending step-degree method to ensure low bias and high diversity. Experimental results on multiple datasets demonstrate superior performance compared to state-of-the-art methods.

**Claims And Evidence:**

The claims presented in the paper are well-supported by both theoretical derivations and experimental validation.

1. The paper rigorously derives the generalization error bound, excess risk bound, and sufficient conditions for the consistency of ensemble clustering. These theoretical results establish a solid foundation for the feasibility and effectiveness of the proposed algorithm, providing strong theoretical support for ensemble clustering method selection.

2. Through comparisons with state-of-the-art methods, the experimental results demonstrate the superiority of the proposed algorithm across multiple datasets. The algorithm obtains good performance in terms of NMI, ARI, and Purity.

3. The paper effectively integrates the bias-diversity tradeoff principle into ensemble clustering optimization. By minimizing bias and maximizing diversity, the proposed approach enhances clustering performance. This concept is further validated through both algorithmic design and empirical results.

**Essential References Not Discussed:**

The paper has cited and discussed relevant prior findings and results necessary for contextualizing its contributions.

**Experimental Designs Or Analyses:**

The experiments effectively demonstrate the advantages of the proposed method, but there are areas that could be further improved for a more comprehensive evaluation.

1. The paper evaluates the algorithm on multiple datasets.

2. The paper discusses key parameters such as convergence rate, number of iterations, and learning rate. However, a more detailed exploration of how these parameters influence performance across different datasets would strengthen the experimental findings.

3. Certain aspects that could enhance the credibility of the results are not explicitly addressed. For instance, ablation studies on the impact of individual components in the algorithm (e.g., the weighting strategy, bias-diversity optimization) could provide deeper insights into the contributions of each part. Additionally, comparisons with a broader range of baseline methods, particularly under different noise conditions, would further support the claims of robustness.

**Methods And Evaluation Criteria:**

In this paper, several evaluation criteria such as NMI, ARI and Purity are adopted at the same time. This comprehensive evaluation method is more comprehensive and can evaluate the performance of the algorithm from different perspectives. NMI and ARI provide an assessment of association with real labels, while Purity focuses more on clustering accuracy. By using the multi-index evaluation method, the effectiveness of the integrated clustering method can be comprehensively measured and its applicability in practical problems can be ensured.

**Other Comments Or Suggestions:**

I have no other comment.

**Other Strengths And Weaknesses:**

**strengths**
This paper provides a theoretical analysis of the generalization performance of ensemble clustering, addressing a key gap in understanding the theoretical foundations of ensemble clustering.

**weaknesses**

1.	While the paper establishes sufficient conditions for clustering consistency, it does not discuss necessary conditions.

2.	Some symbolic definitions for intermediate processes are omitted, which may affect clarity.

3.	In Section 2.2, the notation i in the definition of \bar{A} is not explicitly explained.

4.	The selection of an appropriate threshold is a challenge.

5.	Although the paper mentions a method for setting the threshold, it does not thoroughly analyze the impact of different threshold choices on the results.

6.	The computational complexity of the proposed algorithm is not analyzed in detail.

7.	The optimization process, which uses a reduced gradient descent method to optimize the weighted matrix W and spectral embedding Z, lacks an in-depth discussion of its convergence, efficiency, and stability in practical applications. Since gradient descent methods can be sensitive to initialization, it remains unclear whether the proposed approach guarantees a global optimal solution.

8.	While the paper validates its theoretical findings through experiments, these primarily focus on performance comparisons between algorithms. The verification of the theoretical results, such as experimentally confirming whether the convergence rates of generalization error and excess risk align with theoretical expectations—is not thoroughly addressed.

**Questions For Authors:**

This paper presents a new integrated clustering algorithm based on theoretical framework. How do different hyperparameter Settings affect the performance of the algorithm, and how to choose the optimal hyperparameter?

How do the results of the proposed algorithm compare with other state-of-the-art methods in terms of computational efficiency and scalability?

**Relation To Broader Scientific Literature:**

This paper fills this gap in the generalization performance of ensemble clustering by providing a comprehensive analysis of the generalization error, excessive risk, and consistency of ensemble clustering.

**Theoretical Claims:**

I have reviewed the validity of the proofs for the theoretical claims presented in the paper. The key theorems—3.1 (generalization error bound), 3.2 (excess risk bound), and 3.3 (consistency)—are derived in a detailed and structured manner. The proof methodology is logical and rigorous, leveraging probability theory and statistical consistency principles. Intuitively, the results align with theoretical expectations, and the reasoning appears sound.

---

> ### Author Rebuttal · Authors · 2025-04-01
>
> **We sincerely thanks for all your constructive comments!**
>
> We denote "Experimental Designs Or Analyses" as E, "Questions For Authors" as Q, and "Weakness" as W to save space.
> >**E2 & W7: Global optimal solution**
>
> Our model is a **convex optimization problem of w** (optimization function is convex and equality constraint is affine). Specifically, for our optimization function $J(w)$, we have
> $$J(aw_1+(1-a)w_2)=\max _{Z\in\Gamma}tr((2\tilde{K}+K^{aw_1+(1-a)w_2})ZZ^T)$$
>
> $$=\max _{Z\in \Gamma}tr((2\tilde K+\sum _{t=1}^m{(aw _{1t}+(1-a)w _{2t})^2K^{(t)}})ZZ^T)$$
> $$\le\max _{Z\in\Gamma}tr((2\tilde{K}+\sum _{t=1}^m{(aw _{1t}^{2}+(1-a)w _{2t}^{2} )K^{(t)}})ZZ^T)\quad Given\ a(a-1)\le 0$$
> $$=\max _{Z\in\Gamma}tr((2a\tilde{K}+aK^{w_1}+2(1-a)\tilde{K}+(1-a)K^{w_2})ZZ^T)$$
> $$\le a\max _{Z\in\Gamma}tr((2\tilde{K}+K^{w_1})ZZ^T)+(1-a)\max _{Z\in\Gamma}tr((2\tilde{K}+K^{w_2})ZZ^T)$$
>
> $=aJ(w_1)+(1-a)J(w_2)$,
>
> which means it is convex. Obviously, the constraint $\sum_{t=1}^m w^{(t)}=1$ is affine. Therefore, it can be theoretically demonstrated that our method will achieve the **global optimal value** with different initializations. As verified by repeated experiments on various datasets, with random initializations (see <https://anonymous.4open.science/r/ICML8598/FigA3.pdf>), our algorithm consistently attains the global minimum.
>
> We initialize the learning rate as $\min(0.1,\min_t(w_t/\nabla_t))$ to maintain $w\ge0$. When reaching the minimum loss with above learning rate, we use the Golden search method for finer updates, stopping when $|w_{new}-w_{old}|<0.001$ (i.e., convergence criterion is 0.001). We set the maximum number of iterations to 100, but in practice, the algorithm terminates after just a few dozen iterations.
> >**W4,5 & Q1: Hyperparameter setting**
>
> **Our algorithm has only one hyperparameter**: threshold $\alpha$ and outperforms SOTA methods with $\alpha$ even fixed at 0.1. Besides, In Appendix E.4, Fig 4, we have exhibited the performance across different datasets when the **threshold ranges from 0.1 to 0.9** using grid search (a method consistent with all the baselines). The results show that when the threshold is in {0.1,0.3}, the model achieves better performance, indicating our algorithm is **not sensitive to the hyperparameter**.
> >**E3: Ablation study & Noise situation**
>
> In Appendix E.6, Table 6, we have conducted ablation experiments. It shows that performance declines when either the Bias or Diversity module is removed, indicating both are **important for optimal results**.
>
> As your suggestions, we add more baselines (AAAI24, TKDD23, Inf Fus22, AAAI21) to validate our method under different levels of noise (from level 10% to 90%). The results show that our method **remains the best** in noise condition, which further demonstrate its robustness. The experiments are in <https://anonymous.4open.science/r/ICML8598/TabB12.pdf>
> >**W6 & Q2: Computational efficiency**
>
> The time complexity of each module in our method is as follows:
> - Construct each similarity: $O(n^2)$
> - Calculate \tilde{K}: $O(n^{2.376})$
> - Solve Z: $O(n^2)$
> - Compute reduced gradient: $O(n^2)$
> - Update $w$: $O(m)$
>
> Note that in matrix multiplication and eigenvalue decomposition, we can employ accelerated methods such as Coppersmith-Winograd algorithm. Thus, the time complexity of our method is $O(n^{2.376})$ and most of the baselines have the same time complexity, as they also involve matrix multiplication. We also conducted time cost experiment (which can be seen in <https://anonymous.4open.science/r/ICML8598/FigA4.pdf>), and the results show our method **outperforms these matrix optimization-based methods** in terms of both performance and time cost.
> >**W1,8: Verification of theoretical results and necessary conditions for consistency**
>
> Our convergence rates of generalization error and excess risk are both $O(\sqrt{\log n/m}+1/\sqrt{n})$. **In Sec 6.3, Fig 3, we have demonstrated the convergence rate of the excess risk bound on real dataset.** According to your comments, we conduct experiment on generalization error rate in <https://anonymous.4open.science/r/ICML8598/FigA5.pdf>, and the result shows that it is also consistent with our theory.
>
> We derived the sufficient condition for consistency is $m\gg\log n$, and **it is noteworthy that we are the first to theoretically depict the relationship between clusterings number and sample size**. Additionally, the conclusion that $m\gg \log n$ is a mild condition, indicating that we only need a few base clusterings to satisfy the consistency condition. However, we think the necessary condition is very challenging and we have rarely seen any similar research on this topic. We would like to leave this as our future work.
> >**W2,3: Symbolic definitions and Notations**
>
> As suggested, we will revise the paper to avoid undefined notations. For example, we will correct the typo where $t$ is mistakenly written as $i$ in definition of $\bar{A}$, and add an additional explanation for it.

---

> > ### Comment · Reviewer_kz2j · 2025-04-08
> >
> > The authors have adequately addressed the major concerns raised in the previous review. Based on the improvements and clarifications provided in the rebuttal, I am raising my score to 3.

---

### Official Review · Reviewer_ms8h · 2025-03-08

**Overall Recommendation:** 3

**Summary:**

The ensemble clustering is the problem of combining multiple base clusterings into a more accurate final clustering result. Prior research shows advances of ensemble clustering in practice while the theoretical analysis has fallen behind.

This paper provides the first generalization bound of emsemble clustering and extend their generalization bound into a new algorithm. The authors also conducted experiments to validate their bounds as evidence.

**Claims And Evidence:**

Yes. The generalization bounds have been proved rigorously and experimental results also provide evidence.

**Essential References Not Discussed:**

No

**Experimental Designs Or Analyses:**

The authors have run experiments against a few existing benchmarks. The evaluation method also makes sense to me. Detailed introductions of experiment design is included in appendix.

**Methods And Evaluation Criteria:**

Yes.

**Other Comments Or Suggestions:**

1. On page 2, section 2.2, the definition of $\bar{A}=\frac{1}{m}\sum_{i=1}^m A^{(t)}$ is a typo?
2. On page 2, section 3, the definition of matrix D can not be found anywhere. Moreover the notation $D^{(t)-\frac{1}{2}}$ is very confusing.

**Other Strengths And Weaknesses:**

Strength:  This paper is a rigorous theoretical study of ensemble learning and the results proved are novel .

Weakness:  This paper could certainly benefit from improvements in its writting quality. There is no formal problem definition and I have to read the Fred-Jain paper to understand many concepts. Authors should not assume every reader is an expert in the area of study.

**Questions For Authors:**

1. Can you please provide some insights on the gap condition between the k-th and k+1-th eigenvalues of K^*? Is it a mild condition or is it generally true in practice?

2. The $\sqrt \frac{\log n}{m}$ bound means it requires $m>>\log n$, namely, using more than $\log n$ base clusterings, to guarantee convergence to the ground truth, which can be achieved by a single clustering (say kernel k-means). Moreover, the experiment also show the loss is divergence when $m=\log \log n<\log n$. Is this a "more is less" contradiction?

**Relation To Broader Scientific Literature:**

Given the success of ensemble clustering in practice,  results in this paper can benifit any machine learning research that conducts ensemble clustering as a subroutine.

**Theoretical Claims:**

I have briely read the statement and proofs in appendix. The claims and the way to get there seems correct.

---

> ### Author Rebuttal · Authors · 2025-04-01
>
> **We sincerely thanks for all your constructive comments!**
> >**Weakness 1: Improve writing quality and clarify some concepts**
>
> Thanks for your suggestions! We will make the following changes in the final version of the paper:
> - Provide a more detailed explanation of ensemble clustering and motivation in Introduction
> - Rename Section 2.2 from "Co-Association Matrix" to "Ensemble Clustering" and add a simple flowchart to illustrate the process. (Fig can be seen in <https://anonymous.4open.science/r/ICML8598/FigA1.pdf>)
> - Add Section 2.3 "Problem Definition" to clarify the problem definition
> - In Section 3 "Generalization Performance" we will provide more details for several equations
> - In Section 4 "Key Factors in Ensemble Clustering" we will include a more detailed derivation of Eq. (8) to Eq. (9)
> - Move some important experimental conclusions into Section 6 Experiments instead of Appendix
> - Review the entire paper to avoid undefined symbols and typos
>
> >**Comment 1: A typo in $\bar{A}=1/m\sum_i^m A^{(t)}$**
>
> The correct form is $\bar{A}=1/m\sum _t^m A^{(t)}$, and we will correct it in the final version.
> >**Comment 2: Definition of matrix $D$ and notation $D^{(t)−1/2}$ are confusing**
>
> $D^{(t)}$ is the degree matrix of similarity matrix $A^{(t)}$, which is diagonal and defined as $D_{jj}^{(t)}=\sum_{i=1}^n A_{ij}^{(t)}$ and $D^{(t)-1/2}$ is defined as a diagonal matrix where the diagonal elements are $(D _{jj}^{(t)})^{-1/2}$ and the off-diagonal elements are 0. We will define it in the final version.
> >**Question 1: Is the gap condition between the k-th and k+1-th eigenvalues of $K^\*$ mild or generally true**
>
> It is a mild condition and can be explained in two aspects.
> 1. $K^*$ is regarded as the true similarity (kernel) of the data, and in many papers regarding kernels, they also adopted this assumption, such as [1] and [2].
> 2. On different datasets, we randomly sampled base clusterings and then calculated their means to approximate $K^*$ (since the expectation is unattainable). The results show that in our 1000 experiments, **not once** were the k-th and (k+1)-th eigenvalues equal. The experiment can be seen in <https://anonymous.4open.science/r/ICML8598/FigA2.pdf>
>
> Therefore, it is a **mild condition** and we will incorporate this condition into the General Assumptions section in the final submission of our paper to avoid misunderstandings.
>
> [1] Error bounds for kernel-based approximations of the Koopman operator.
>
> [2] Scalable Multiple Kernel Clustering: Learning Clustering Structure from Expectation.
>
> >**Question 2: A single clustering algorithm (like kernel k-means) can ensure convergence to the true value, but why does ensemble clustering require $m\gg \log n$. Experiment shows the loss is divergence when $m=\log \log n < \log n$. Is this a "more is less" contradiction?**
>
> - It's important to note that in the generalization analysis of single clustering algorithm (like kernel k-means), it is often assumed that the data features are accessible or the kernel function represents the true similarity relationships in the data. Their studies concern the necessity of generalizing from finite-dimensional matrices to infinite-dimensional integral operators. But in ensemble clustering, our similarity matrices are binary and generated by $n\times 1$ vectors (**no features or kernel functions**). We can view features or kernel functions as high-dimensional representations of the data, while considering the base clusterings in ensemble clustering as **special discrete one-dimensional projections**. The problem we are addressing is whether we can achieve the same results using **only** these discrete one-dimensional embeddings, instead of relying on high-dimensional representations. Our research shows that consistent results can be obtained when $m\gg \log n$. We believe this is valuable not only in ensemble clustering but also in other areas of machine learning.
>
> - Note that the loss diverges when $m = \log \log n$ in Section 6.3, Figure 3. However, in this experiment, the sample size $n$ increases with $m$, **rather than being fixed**. In Appendix E.6, Figure 5, we present an experiment where the clusterings size $m$ is increased while the sample size $n$ is fixed. This experiment shows that our clustering accuracy improves as $m$ increases (Our convergence rate $O(\sqrt{\frac{\log n}{m}} + 1/\sqrt{n})$ also shows that as 𝑚 increases and $n$ is fixed, the error is reduced). A more intuitive explanation is that, in ensemble clustering, we would like to approximate the true kernel function using $m$ binary similarity matrices (with dimension $n$). Here, $n$ should be considered as the **feature dimension** of the matrices, and $m$ as the **number of samples** (number of similarity matrices). As the feature dimension $n$ increases, we need to add more samples (similarity matrix) to avoid underfitting. Thus, this is **not a "more is less contradiction"**.

---

### Decision · Program_Chairs · 2025-05-01

**Decision:**

Accept (poster)

**Comment:**

All reviewers recommend acceptance, and appreciate the theoretical contributions of the work. Although there were some technical queries raised in the initial reviewing stage, the authors did a good job in addressing these.
In addition to some strong theoretical results in a field which is not brimming with such contributions, the proposed methodology also shows strong practical performance in comparison with relevant benchmarks.